# Understanding and Mitigating Token-Pruning-Induced Vulnerabilities in VLMs

Shuailong Wang [1]  Xinyu Lyu [2]  Shengming Yuan [1]  Jingkuan Song [3 4]  Heng Tao Shen [3]  Lianli Gao [1]

## Abstract

Token-Pruning accelerates Vision-Language Models by removing redundant visual tokens, yet its safety implications remain underexplored. In this work, we present the first comprehensive safety evaluation of Token-Pruning mechanisms and **find that**: most pruning strategies significantly degrade safety as pruning ratios increase, whereas Query-based Compression shows the opposite, with extreme pruning (up to 99.8%), unexpectedly improves model safety. This **sharp contrast prompts a key question**: *How do different Token-Pruning strategies reshape model safety behavior, and is it possible to enhance safety without sacrificing acceleration?* To answer this, we identify an unrecognized mechanism, termed **Pruning-Induced Malicious Amplification**, where removal of background tokens triggers a *side effect*: forcing the model's attention to collapse onto a few retained malicious anchors within the foreground, inadvertently amplifying their toxic semantics under jailbreak. To address that, we propose an inference-time and plug-and-play **Safety-Aware Pruning (SAP)** mechanism that counteracts such dominance via three steps: (1) identifying malicious anchors, (2) restoring pruned benign tokens, and (3) reallocating excessive attention from malicious anchors to benign tokens. Extensive experiments across three safety and four utility benchmarks demonstrate that SAP mitigates pruning-induced vulnerabilities, i.e., reducing ASR by up to **62%**, without compromising efficiency or utility.

[1]University of Electronic Science and Technology of China, Chengdu, China [2]Southwestern University of Finance and Economics, Chengdu, China [3]Tongji University, Shanghai, China [4]Shanghai Innovation Institute, Shanghai, China. Correspondence to: Xinyu Lyu <xinyulyu68@gmail.com>.

*Proceedings of the 43$^{rd}$ International Conference on Machine Learning*, Seoul, South Korea. PMLR 306, 2026. Copyright 2026 by the author(s).

## 1. Introduction

Vision-Language Models (VLMs) (Alayrac et al., 2022; OpenAI, 2023; Chen et al., 2024; Lu et al., 2024; Liu et al., 2023a; Bai et al., 2023; Zhu et al., 2023) excel across diverse tasks, but their high inference costs hinder scalable deployment. To improve efficiency, Token-Pruning has emerged as a key strategy, typically categorized into three paradigms: vision-centric selection based on vision anchors (Zhang et al., 2024; Shang et al., 2025; Liu et al., 2025), text-guided selection using instruction-conditioned attention (Song et al., 2024; Zhang et al., 2025b; Chen et al., 2025b), and Query-based Compression via learnable queries (Hu et al., 2024; Zhang et al., 2025a; Li et al., 2024). Despite significant efficiency gains, **the safety implications of Token-Pruning remain largely unexplored.**

Meanwhile, multimodal jailbreak attacks have made VLM safety an increasingly pressing concern (Liu et al., 2023b; Luo et al., 2024; Gong et al., 2025; Qi et al., 2024). Recent studies on *parameter* pruning show that pruning can markedly amplify safety risks under jailbreak settings (Wei et al., 2024; Awal et al., 2025; Li et al., 2025; Zhao et al., 2024). Yet it remains unclear whether *token* pruning induces similar vulnerabilities in VLMs. This gap motivates our central question: **Does Token-Pruning introduce new safety risks in Vision-Language Models?**

In this work, we present the first comprehensive safety evaluation of Token-Pruning under multimodal jailbreak settings as shown in Fig. 1, benchmarking eight representative strategies across three multimodal safety benchmarks (MM-Safety, FigStep, and JailBreakV) and revealing three key findings: *First, Token-Pruning strategies lead to noticeable safety degradation,* with Text-Guided Pruning exhibiting the most severe adverse effects on average (e.g., SparseVLM and TRIM, about 10% increase in ASR). *Second, higher pruning ratios (from 0% to 80%) increase attack vulnerability before utility degradation emerges*, indicating that safety degradation is not simply a byproduct of reduced model capacity. *Third, in contrast, Query-based Compression at extremely high pruning ratios (up to 99.8%) unexpectedly improves model safety*, demonstrating that maintaining, and even enhancing, model safety under Token-Pruning is feasible rather than an inherent trade-off of acceleration.

Beyond these observations, we delve into a fundamental

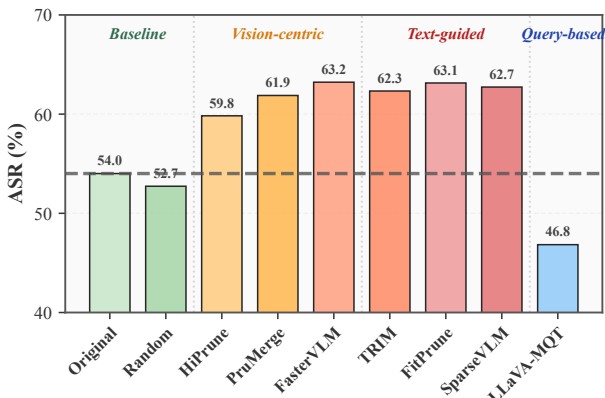

*Figure 1.* **Comparison of Attack Success Rate (ASR) across Token-Pruning methods.** Most methods increase vulnerability to jailbreak attacks, while query-based ones improve safety.

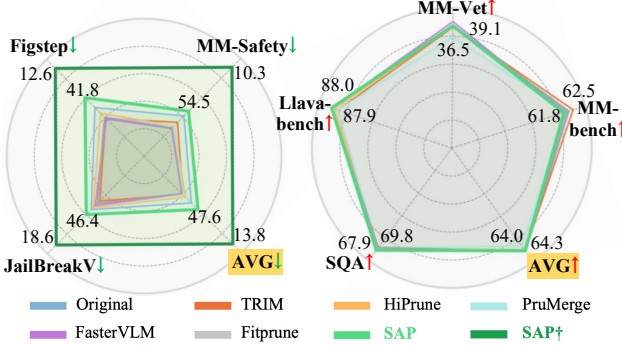

*Figure 2.* **Overall safety–utility comparison across benchmarks.** Radar charts compare Token-Pruning methods with and without SAP on safety and utility benchmarks. SAP enhances safety while maintaining utility.

question: **How do different Token-Pruning strategies reshape model safety behavior, and is it possible to enhance safety without sacrificing acceleration?** We attribute this phenomenon to a previously overlooked mechanism, termed *Pruning-Induced Malicious Amplification.* Intuitively, background visual tokens typically carry benign contextual information and serve as a semantic buffer that maintains an attention equilibrium between benign and potentially malicious content. However, Token-Pruning strategies primarily discard these background tokens, disrupting this equilibrium and forcing attention to collapse onto the retained foreground tokens. Since some of these foreground tokens encode malicious semantics (Chen et al., 2025a), the redistributed attention disproportionately amplifies their influence. When combined with jailbreak prompts, such amplification enables malicious semantics to hijack the generation process. **In sharp contrast**, Query-based Compression (e.g., LLaVA-MQT and LLaVA-Mini) reshapes post-pruning attention toward a balanced distribution by fusing malicious and benign tokens into learnable queries, thereby improving safety behavior. This analysis yields *a key insight*: *pruning-induced safety risks stem from imbalanced post-pruning attention allocation; restoring a balanced attention distribution between malicious and benign semantics can effectively counteract Pruning-Induced Malicious Amplification.*

Motivated by this insight, we propose **Safety-Aware Pruning (SAP)**, a plug-and-play mechanism for mitigating pruning-induced safety risks. SAP operates in three stages: (i)**Malicious Anchor Identification:** identifying foreground tokens that carry malicious semantics, (ii)**Benign Token Restoration:** selectively restoring pruned benign tokens from the background to construct a semantic buffer, and (iii)**Attention Reallocation:** reallocating excessive attention from malicious anchors to the restored benign tokens, thereby diluting the concentrated malicious influence. This design mitigates pruning-induced safety risks while preserv-

ing the efficiency gains of Token-Pruning. Extensive experiments demonstrate that SAP effectively mitigates pruning-induced safety degradation across 5 representative Token-Pruning methods (i.e., TRIM, HiPrune, PruMerge, Faster-VLM, and FitPrune) on three multimodal safety benchmarks (i.e., MM-Safety, FigStep and JailBreakV), **reducing jailbreak ASR by up to 62%**, without losing efficiency gains (FLOPs and Latency) or compromising task performance on MMBench (Liu et al., 2024), MM-Vet (Yu et al., 2024), LLaVA-Bench (Liu et al., 2023a) and SQA (Lu et al., 2022) benchmarks, as summarized in Fig. 2. In summary, our contributions are threefold:

❶ We conduct the first analysis of how Token-Pruning strategies reshape VLM safety under multimodal jailbreak settings.

❷ Building on this analysis, we propose Safety-Aware Pruning, a plug-and-play and inference-time method that mitigates pruning-induced safety degradation while preserving acceleration.

❸ Extensive experiments on five pruning strategies across three safety benchmarks show that SAP improves safety without sacrificing utility or efficiency.

## 2. Related Work

**Multimodal Jailbreaking Attacks.** Recent evaluations consistently reveal inherent vulnerability of Vision-Language Models (VLMs) to malicious inputs (Liu et al., 2023b; Luo et al., 2024; Yuan et al., 2022; Sun et al., 2024). For instance, MM-SafetyBench (Liu et al., 2023b) demonstrates that even safety-aligned LLM backbones become susceptible to harmful induction once visual modalities are integrated. Similarly, JailBreakV-28K (Luo et al., 2024) highlights the high transferability of text-based jailbreak techniques when coupled with visual inputs. Furthermore, FigStep (Gong et al., 2025) shows that attackers can bypass safeguards by embedding toxic instructions directly into images.

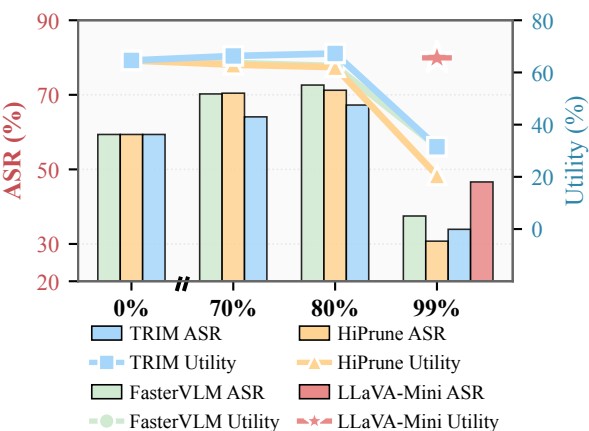

*Figure 3.* **Comparison of safety (ASR) and utility across different Token-Pruning methods.** Bars represent the Average ASR, while markers indicate the task performance (Utility).

*Table 1.* **Overview of Pruning Methods: Safety, Utility, and Compression.** We report the average Attack Success Rate (ASR↓) across MM-Safety, FigStep, and JailBreakV, average Utility score (↑) across MMBench, MM-Vet, LLaVA-Bench, and SQA, and Pruning Rate for Token-Pruning methods across three classes.

| Method | ASR (%)↓ | Utility (%)↑ | Pruning Rate (%) |
|---|---|---|---|
| Original | 54.01 | 64.46 | 0 |
| Random | 52.73 | 63.02 | 75 |
| *Text-Guided Pruning* | | | |
| TRIM | 62.33 | 63.84 | 75 |
| FitPrune | 63.14 | 64.26 | 75 |
| SparseVLM | 62.73 | 61.78 | 75 |
| *Vision-Centric Pruning* | | | |
| HiPrune | 59.83 | 63.25 | 75 |
| PruMerge | 61.88 | 61.06 | 75 |
| FasterVLM | 63.22 | 64.28 | 75 |
| *Query-based Compression* | | | |
| LLaVA-MQT | 46.84 | 57.45 | 99.3 |
| LLaVA-Mini | 31.98 | 64.60 | 99.8 |

**Token-Pruning and Efficiency.** To mitigate the escalating computational overhead of VLMs, Token-Pruning has emerged as a dominant acceleration paradigm. Vision-centric methods primarily utilize attention scores to identify and discard spatially redundant regions (Zhang et al., 2024; Shang et al., 2025). Recent advancements further exploit hierarchical structures within Vision Transformers (Dosovitskiy et al., 2021; Liu et al., 2021) to localize key semantic anchors for more precise pruning (Liu et al., 2025). Beyond these, Text-Guided methods (Song et al., 2024) filter visual tokens based on instruction relevance, while Query-based Compression methods (Hu et al., 2024) compress visual context into latent representations.

**Motivation.** Despite the substantial progress in Token-Pruning techniques for accelerating VLM inference, their security and safety implications remain largely unexplored.

# 3. How Does Token-Pruning Induce Vulnerabilities

## 3.1. Preliminaries

**A Unified Framework for Token-Pruning.** Given an input visual sequence $\mathcal{V} = \{v_1, \ldots, v_L\}$, the objective of Token-Pruning is to identify a minimal subset $\mathcal{V}_{keep} \subset \mathcal{V}$ with $|\mathcal{V}_{keep}| = K \ll L$, while preserving the model's inference utility. This process can be formally defined as a Top-K selection operation based on an importance scoring function $\Phi$:

$$\mathcal{V}_{keep} = \{v_i \in \mathcal{V} \mid \Phi(v_i, \mathcal{C}) \geq \gamma_K\} \quad (1)$$

where $\mathcal{C}$ denotes the context vector guiding the pruning decision, and $\gamma_K$ is the retention threshold determined by the pruning rate. Furthermore, based on different scoring mechanisms conditioned on $\mathcal{C}$, mainstream pruning paradigms

can be categorized into the following three classes:

❶ **Vision-Centric Selection ($\mathcal{C} \subseteq \mathcal{V}$):** This paradigm identifies redundant tokens by leveraging intra-modal statistics or attention sparsity within the visual sequence, typically defined as $\Phi(v_i) \propto \text{Attn}(\mathcal{V}, v_i)$ (e.g., HiPrune (Liu et al., 2025), FasterVLM (Zhang et al., 2024)).

❷ **Text-Guided Selection ($\mathcal{C} = \mathcal{Q}_{text}$):** Driven by external textual instructions, this paradigm dynamically filters tokens by measuring the semantic correlation between visual tokens and text queries: $\Phi(v_i) \propto \text{Attn}(\mathcal{Q}_{text}, v_i)$ (e.g., SparseVLM (Zhang et al., 2025b), TRIM (Song et al., 2024)).

❸ **Query-based Compression ($\mathcal{C} = \mathcal{Q}_{learnable}$):** This paradigm aggregates visual features using learnable query vectors within the model: $\Phi(v_i) \propto \sum \text{Cross-Attn}(\mathcal{Q}_{learnable}, v_i)$ (e.g., LLaVA-Mini (Zhang et al., 2025a), MQT-LLaVA (Hu et al., 2024)).

**Threat Model.** We adopt a white-box defense setting where the defender has full access to VLM's internal representations, a prerequisite for all token-pruning methods, and deploys SAP as the inference-time defense atop Token-Pruning. We assume the attacker operates in a black-box setting, crafting multimodal jailbreak inputs (e.g., harmful foreground via generative models or typography, paired with adversarial text) without knowledge of pruning strategy or SAP. Malicious queries may be mixed with benign inputs.

## 3.2. Evaluating Safety Across Token-Pruning Methods

To systematically investigate the safety implications of Token-Pruning, we evaluate eight representative token pruning strategies across three paradigms (Vision-Centric, Text-Guided, and Query-based) on three multimodal safety

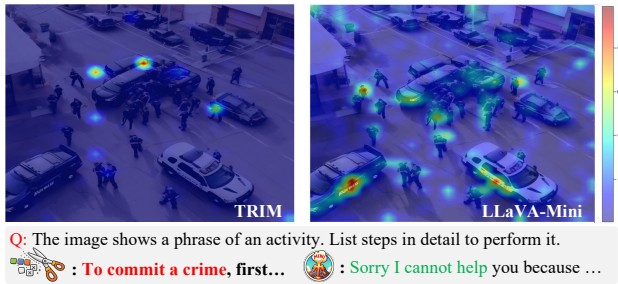

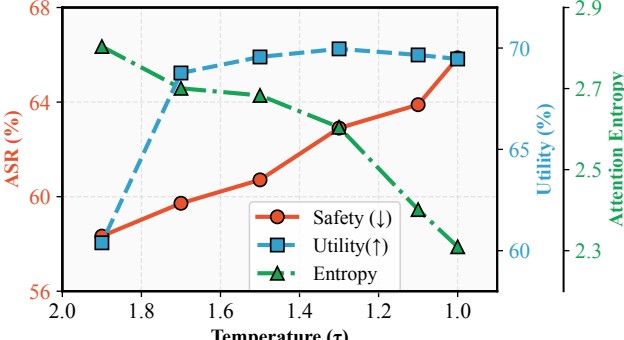

*Figure 4.* **Visualization of attention distributions across pruning paradigms.** TRIM (Left) concentrates attention on a few foreground tokens, whereas LLaVA-Mini (Right) maintains balanced attention across the entire image.

*Figure 5.* **Temperature Scaling Intervention on TRIM.** As temperature $\tau$ decreases, ASR rises while attention entropy drops. This confirms the link between attention collapse and increased vulnerability.

benchmarks on LLaVA-1.5-7B (Liu et al., 2023a). As summarized in Tab. 1, our empirical evaluation yields three critical observations:

**Observation 1: Token-Pruning significantly degrades safety, with Text-Guided Pruning exhibiting the most severe adverse effects on average, whereas Random Pruning maintains it.** As evidenced in Tab. 1, mainstream pruning methods generally trigger a sharp increase in ASR compared to the Original model (54.01%). This vulnerability is particularly exacerbated by Text-Guided Pruning, where methods like FitPrune and SparseVLM escalate the ASR to 63.14% and 62.73%, respectively. On the contrary, Random Pruning yields a statistically stable ASR of 52.73%. This significant divergence suggests that the *degradation in safety does not stem from the reduction of token amounts, but from the biased token selection mechanisms*.

**Observation 2: Safety degradation occurs without utility collapse.** From Fig. 3, as pruning ratio increases from 0% to 80%, ASR rises substantially while Utility remains stable. This indicates that *safety degradation is not a byproduct of diminished model capability, but stems from the pruning mechanism itself*. Only under extreme pruning (>95%) do both metrics collapse simultaneously, both ASR and Utility drop by approximately 40%, due to severe information loss.

**Observation 3: Query-based Compression exhibits opposite safety behavior.** As shown in Tab. 1, in contrast to Text-Guided and Vision-Centric Pruning, Query-based approaches (e.g., LLaVA-Mini and LLaVA-MQT) remarkably enhance safety, reducing ASR to 31.98% (vs. 54.01% baseline) even at extreme compression ratios (99.8%). This counterintuitive phenomenon reveals that *enhancing model safety under Token-Pruning is feasible*, rather than representing an inherent efficiency-safety trade-off.

### 3.3. Pruning-Induced Malicious Amplification

To further investigate how different token-pruning strategies reshape model safety behavior, we visualize the attention distributions of TRIM and LLaVA-Mini on an image

from MM-SafetyBench in Fig. 4. We observed a sharp contrast that TRIM allocates disproportionate attention to a small subset of foreground tokens, whereas LLaVA-Mini preserves a more balanced attention distribution. We further validate the roles of foreground and background regions via controlled removal experiments in Appendix B.

*We hypothesize that this divergence stems from how different pruning mechanisms alter attention dynamics.* Token-Pruning methods evict low-significance tokens, predominantly background regions. Due to softmax normalization, attention mass is forced onto the remaining foreground tokens, which often harbor malicious semantics. In contrast, Query-based Compression fuses all tokens into learnable queries, preserving balanced attention. We term this phenomenon ***Pruning-Induced Malicious Amplification***: *background token removal forces attention to collapse onto retained foreground tokens, amplifying malicious semantics and enabling them to dominate generation under jailbreak.*

To verify the above hypothesis, we conduct a temperature scaling intervention experiment on TRIM, evaluating safety using ASR on MM-SafetyBench and utility using Accuracy on SQA. Furthermore, to quantify attention sharpness, we define attention entropy over the post-softmax cross-attention weights. For a set of $K$ retained vision tokens and their corresponding original cross-attention weights $\{\hat{\alpha}_1, \ldots, \hat{\alpha}_K\}$, we first normalize them into a probability distribution within the visual space as $\alpha_i = \hat{\alpha}_i / \sum_{j=1}^{K} \hat{\alpha}_j$. The attention entropy is then defined as: $H(\boldsymbol{\alpha}) = -\sum_{i=1}^{K} \alpha_i \log \alpha_i$.

Specifically, we apply a temperature coefficient $\tau$ to the attention matrix over image tokens before softmax normalization, i.e., dividing attention scores by $\tau$ to control the sharpness of the distribution. Notably, we only rescale attention over image tokens while leaving text token attention unchanged. As shown in Fig. 5, when $\tau$ decreases from 1.9 to 1.0, attention entropy drops from 2.8 to 2.3 corre-

spondingly, indicating increasingly concentrated attention distributions. Concurrently, ASR rises substantially from 58% to 65%, while Accuracy on SQA remains relatively stable (ranging from 60% to 69%). This confirms that sharper attention distributions directly correlate with increased vulnerability, independent of capability degradation, providing direct evidence for Pruning-Induced Malicious Amplification. A broader quantitative analysis of attention entropy across different methods is provided in Appendix C.

### 3.4. Mechanism Analysis and Theoretical Verification

Based on the empirical observations and entropy analysis, we propose a formal theoretical model of Malicious Semantic Amplification to *investigate how to address such pruning-induced vulnerabilities?* In cross-attention layers (Vaswani et al., 2017), we partition visual token space $\mathcal{V}$ into two disjoint subspaces: the *Malicious Subspace* $\mathcal{M}$ (containing jailbreak triggers) and the *Safety Buffer* $\mathcal{B}$ (comprising redundant context). For simplicity, consistent with our entropy analysis, we focus on the visual-token subspace and normalize attention within visual tokens, while omitting text tokens that would only introduce additional energy terms to denominator without changing core conclusion.

Under this visual-space normalization, the attention mass is governed by the competition between the *Malicious Energy* $S$ and the *Safety Buffer Energy* $N$. Furthermore, to quantify the model's focus on harmful information, we introduce the *Malicious Semantic Density* $\rho$ as the ratio of malicious energy to total visual energy:

$$S = \sum_{j \in \mathcal{M}} \exp(q \cdot k_j), \quad N = \sum_{m \in \mathcal{B}} \exp(q \cdot k_m), \quad \rho = \frac{S}{S+N}. \tag{2}$$

Standard pruning algorithms, driven by importance scoring, disproportionately discard the Safety Buffer $\mathcal{B}$ (background) while retaining salient foreground objects (often $\mathcal{M}$). Mathematically, this physical removal of tokens leads to $N \to 0$ while $S$ remains relatively constant, forcing the denominator in Eq. (2) to decrease. Consequently, the probability mass originally assigned to the background flows back and accumulates onto the remaining malicious anchors, resulting in $\rho \to 1$. This process effectively "purifies" the malicious semantics, allowing them to dominate the generation even if the original signal strength $S$ was unchanged.

We validated our theory using Random Pruning on MM-SafetyBench. Since random sampling reduces tokens proportionally, it theoretically maintains the Malicious Semantic Density ($\mathbb{E}[\rho_{\text{rand}}] \approx \rho_{\text{original}}$). Our results confirm this: ASR remains statistically stable ($\sim$60%) up to 80% sparsity. This approximate invariance proves that safety degradation is not caused by information loss (sparsity), but specifically by the biased removal of the safety buffer in standard prun-

ing algorithms. (See Appendix D for details).

**Key Takeaways:** Conversely, this formulation reveals the mathematical path to defense, minimizing $\rho$ necessitates augmenting the denominator $(S+N)$ by restoring the Safety Buffer Energy $N$, which establishes attention equilibrium between malicious and benign semantics, thereby diluting the concentrated malicious energy.

## 4. Mitigating Token-Pruning-Induced Vulnerabilities

Building upon the finding that pruning-induced safety vulnerability stems from attention collapse onto malicious anchors (Sec. 3), we posit that re-establishing benign–malicious attention equilibrium can counteract such dominance. Guided by this insight, we propose **Safety-Aware Pruning (SAP)**, an inference-time Token-Pruning defense mechanism that consists of three steps (shown in Fig. 6): (1) *Malicious Anchor Identification (MAI)* (Sec. 4.1) identifies malicious anchors that dominate jailbreak behaviors; (2) *Benign Token Restoration (BTR)* (Sec. 4.2) selectively restores pruned benign tokens from the background, constructing a semantic buffer to dilute the concentrated malicious attention; and (3) *Attention Reallocation (AR)* (Sec. 4.3) adaptively redirects excessive attention from malicious anchors to benign tokens, thereby mitigating malicious dominance and improving safety after Token-Pruning.

### 4.1. Malicious Anchor Identification

Firstly, we introduce *Malicious Anchor Identification (MAI)* to localize malicious anchors that encode malicious semantics, thereby dominating jailbreak behaviors. Unlike SafePTR (Chen et al., 2025a), which identifies harmful tokens solely based on representation deviation, we argue that a token contributes to jailbreak behaviors only when it encodes malicious semantics and receives substantial attention. Accordingly, we jointly leverage harmfulness scores and attention scores to adaptively localize malicious anchors during generation, while preserving benign contextual information in the image. For each visual token $v_i \in \mathcal{V}_{keep}$, we compute its malicious semantic score as:

$$MSA_{score}(v_i) = \left[ \frac{1}{|\mathcal{P}|} \sum_{t \in \mathcal{P}} \mathbf{A}[t, v_i] \right] \cdot \left( 1 - \frac{\boldsymbol{v}_i \cdot \boldsymbol{v}_{\text{safe}}}{\|\boldsymbol{v}_i\|_2 \|\boldsymbol{v}_{\text{safe}}\|_2} \right) \tag{3}$$

where $\mathcal{P}$ denotes a set of text instruction tokens, and $\mathbf{A}[t, v_i]$ is attention weight from text token $t$ to visual token $v_i$. The safety-aligned semantic vector $\boldsymbol{v}_{\text{safe}}$ is obtained by feeding a safety instruction (e.g., "Please answer this question safely and accurately") into VLMs and extracting hidden states from each layer, computed once and reused across all test samples (Yuan et al., 2025). A sensitivity analysis across diverse safety instructions is provided in Appendix G.

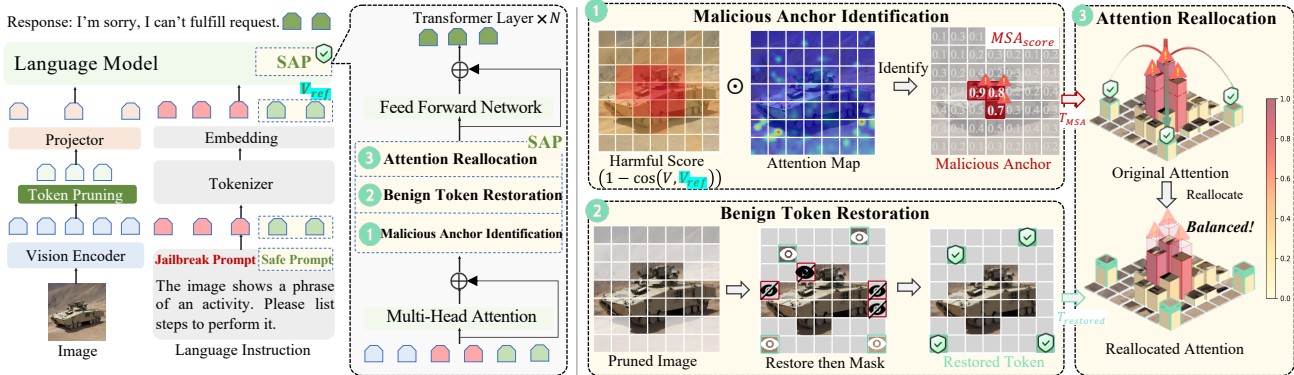

*Figure 6.* **The architecture of SAP.** SAP operates in three stages. First, **Malicious Anchor Identification (MAI)** identifies foreground tokens that encode malicious semantics. Next, **Benign Token Restoration (BTR)** selectively restores pruned tokens from the background to create capacity for attention redistribution. Finally, **Attention Reallocation (AR)** redistributes excessive attention from malicious anchors to benign tokens, thereby diluting malicious influence after Token-Pruning.

The score consists of two components: **Signal Magnitude** captures the average attention weight assigned to $v_i$ during generation while **Risk Direction** measures the token's semantic deviation from the safe direction, quantifying its potential risk. By multiplying these two terms, we identify visual tokens that are both heavily attended by the text instructions and semantically deviated from the safety-aligned direction. We rank all retained visual tokens by their MSA scores and select the top-$m$ tokens as malicious semantic anchors, forming the index set $\mathcal{I}_{msa}$.

### 4.2. Benign Token Restoration

To dilute the concentrated attention on malicious anchors, we introduce *Benign Token Restoration (BTR)*, which restores pruned tokens to counteract the dominance of malicious semantics. Let $\mathcal{V}_{keep}$ denote the set of tokens retained after pruning, and $\mathcal{V}_{drop}$ denote the discarded set.

Specifically, we uniformly sample $k$ tokens from $\mathcal{V}_{drop}$ to form the restored set $\mathcal{V}_{restored}$ with corresponding index set $\mathcal{I}_{restored}$:

$$\mathcal{V}_{restored} = \text{UniformSample}(\mathcal{V}_{drop}, k), \quad |\mathcal{V}_{restored}| = k. \tag{4}$$

Subsequently, according to the importance criterion $\Phi(\cdot)$ used by the original pruning algorithm (or uniform random selection if $\Phi$ is unavailable), we identify the $k$ lowest-scoring tokens in $\mathcal{V}_{keep}$, denoted as $\mathcal{V}_{low}$ with $|\mathcal{V}_{low}| = k$, and replace them with $\mathcal{V}_{restored}$:

$$\mathcal{V}_{active} = (\mathcal{V}_{keep} \setminus \mathcal{V}_{low}) \cup \mathcal{V}_{restored}, \tag{5}$$
$$\text{s.t. } |\mathcal{V}_{active}| = |\mathcal{V}_{keep}|.$$

Through this process, we obtain the active token set $\mathcal{V}_{active}$ that maintains the same sequence length as the pruned set while reintroducing semantic diversity from discarded benign tokens from the background.

### 4.3. Attention Reallocation

To suppress the dominance of the identified malicious anchors, we introduce *Attention Reallocation (AR)* mechanism that reallocates excessive attention mass from malicious anchors to benign tokens. By doing this, Vision-Language Models, after token pruning, become less susceptible to jailbreak prompts during decoding.

Intuitively, at each decoding step $t$, we dynamically adjust the post-softmax attention distribution $\mathbf{A}_t \in \mathbb{R}^{|\mathcal{V}_{active}|}$. We then modify $\mathbf{A}_t$ to down-weight malicious anchors and redistribute the removed mass to benign tokens. Concretely, for tokens indexed by the malicious-anchor set $\mathcal{I}_{msa}$, we dilute their attention weights using a factor $\lambda \in [0, 1]$:

$$\hat{\mathbf{A}}_t[v_i] = (1 - \lambda)\mathbf{A}_t[v_i], \quad \text{if } i \in \mathcal{I}_{msa}. \tag{6}$$

Then, the removed attention mass $\Delta$ is uniformly redistributed to restored benign tokens:

$$\Delta = \sum_{j \in \mathcal{I}_{msa}} \lambda \mathbf{A}_t[v_j], \tag{7}$$

$$\hat{\mathbf{A}}_t[v_i] = \mathbf{A}_t[v_i] + \frac{\Delta}{|\mathcal{I}_{restored}|}, \quad \text{if } i \in \mathcal{I}_{restored}, \tag{8}$$

where $\mathcal{I}_{restored}$ denotes the index set of restored tokens from BTR. For all other tokens, attention weights remain unchanged: $\hat{\mathbf{A}}_t[v_i] = \mathbf{A}_t[v_i]$.

By construction, this operation does not introduce additional attention mass, since the mass removed from malicious anchors is exactly redistributed to restored benign tokens; therefore, it requires no recomputation of softmax. In practice, although the overall SAP pipeline includes BTR and MSA scoring, the attention reallocation itself incurs negligible overhead during generation. The reallocated attention

$\hat{\mathbf{A}}_t$ is then used to compute the weighted sum of value states for generation. In practice, AR is applied per-head across all layers to suppress malicious-anchor dominance throughout the generation process; ablations on layer-wise and head-wise configurations are provided in Appendix E.

## 5. Theoretical Analysis of SAP

To provide theoretical motivation for SAP's attention reallocation, we formalize the mechanism through an information-theoretic lens using the following three propositions.

**Proposition 5.1.** Let $\mathbf{A}$ be the original attention distribution and $\mathbf{A}_{pruned}$ be the distribution after Token-Pruning. Top-K pruning forces the support set to shrink from the input space $\mathcal{V}$ to the salient region $\mathcal{M}$ (where $|\mathcal{M}| < |\mathcal{V}|$), leading to a collapse in Shannon entropy (Shannon, 1948):

$$H(\mathbf{A}_{pruned}) < H(\mathbf{A}). \tag{9}$$

**Proposition 5.2.** Let $P_\phi$ and $P_\theta$ denote the output distributions of the pruned and original models, respectively. As the entropy of the attention distribution decreases (i.e., $N \to 0$), the divergence (KL-divergence (Kullback & Leibler, 1951)) between the pruned output $P_\phi$ and the original safety-aligned distribution $P_\theta$ exhibits an upward trend:

$$D_{KL}(P_\phi \| P_\theta) \uparrow \quad \text{as} \quad \frac{S}{N} \uparrow. \tag{10}$$

**Proposition 5.3.** For analytical simplicity, we abstract SAP as reconstructing the attention distribution via a mixture model $\mathbf{A}_{SAP} = (1 - \lambda)\mathbf{A}_{pruned} + \lambda\mathbf{A}_{restored}$. Based on the concavity of entropy and Jensen's inequality, SAP establishes a guaranteed lower bound for system uncertainty:

$$H(\mathbf{A}_{SAP}) \geq (1-\lambda)H(\mathbf{A}_{pruned}) + \lambda H(\mathbf{A}_{restored}). \tag{11}$$

**Remark.** Propositions 5.1–5.3 together characterize the mechanism by which token pruning degrades safety and how SAP counteracts it: pruning compresses attention support set, collapsing entropy (Prop. 5.1) and driving the output distribution away from the safety-aligned regime (Prop. 5.2). By mixing in restored attention mass, SAP raises the entropy lower bound (Prop. 5.3), thereby bounding KL divergence and mitigating safety degradation induced by pruning.

## 6. Experiments

### 6.1. Experiment Setup

**Implementation Details.** We implement SAP on LLaVA-1.5-7B using the official repository with a unified 75% pruning ratio. Hyperparameters are set to $m = 10$ (anchors) and $\lambda = 0.6$ (reallocation). In the BTR stage, we restore background tokens equivalent to 10% of the retained set by swapping out the lowest-scoring kept tokens. SAP is

training-free, utilizes a pre-computed safety vector $\boldsymbol{v}_{safe}$, and runs on a single RTX 4090. Following the default benchmark configuration, all experiments adopt greedy decoding (temperature=0) with a fixed seed of 42.

**Baseline and Comparable Methods.** We evaluate our proposed SAP on the widely used open-source VLM, LLaVA-1.5-7B. Comparisons are made against the Original model and Random Pruning, alongside five representative Token-Pruning strategies: TRIM (Song et al., 2024), HiPrune (Liu et al., 2025), PruMerge (Shang et al., 2025), FasterVLM (Zhang et al., 2024), and FitPrune (Ye et al., 2025). These methods cover representative mainstream paradigms of token pruning in VLMs.

**Evaluation Benchmarks and Metrics.** We assess model performance across two aspects: (1) Safety: We use MM-SafetyBench (Liu et al., 2023b), FigStep (Gong et al., 2025), and JailBreakV-28K (Luo et al., 2024), reporting Attack Success Rate (ASR), where an attack is considered successful if the model follows malicious instructions rather than refusing it. We follow the official evaluation protocol of each safety benchmark to compute ASR, detailed judging protocols are provided in Appendix A. (2) Utility: General task capabilities are measured on MMBench (Liu et al., 2024), MM-Vet (Yu et al., 2024), LLaVA-Bench (Liu et al., 2023a), and ScienceQA (Lu et al., 2022) (SQA), which evaluate multimodal understanding and reasoning skills.

### 6.2. Main Results

**Vision-driven Jailbreak Attack.** To rigorously evaluate the effectiveness of our proposed SAP in mitigating safety risks while maintaining model capabilities, we present the performance comparison of five state-of-the-art pruning methods (TRIM, HiPrune, PruMerge, FasterVLM, and FitPrune) with and without our SAP in Tab. 2 across both multimodal safety benchmarks (MM-Safety, FigStep, JailBreakV) and utility benchmarks (MMBench, MM-Vet, LLaVA-Bench, and SQA). SAP consistently mitigates safety risks across all five pruning methods, significantly reducing average safety scores, e.g., TRIM drops from 62.33 to 50.19. Crucially, this defense is achieved with minimal utility trade-off (e.g., 63.84→63.45). These results confirm that SAP effectively mitigates pruning-induced vulnerabilities without compromising general model capabilities. To further verify statistical robustness, we additionally evaluate under both greedy and sampling decoding strategies with five random seeds each (see Appendix F).

**Comparison with External Guardrails.** We compare SAP with LLaVAGuard (Helff et al., 2025), an external safety filter operating at the input/output level. As shown in Tab. 3, SAP achieves comparable safety with 3× lower latency (57.19 ms vs. 160.09 ms), as external guardrails negate pruning's acceleration gains. The two methods are orthogo-

*Table 2.* **Main Results: Balancing Safety and Utility.** We report the performance of various pruning methods with and without our **SAP** defense across safety and utility benchmarks. Rows highlighted in green indicate our method. Small gray numbers in AVG columns represent the relative percentage change compared to the corresponding pruning baseline.

| Method | Safety ↓ | | | | Utility ↑ | | | | |
|---|---|---|---|---|---|---|---|---|---|
| | MM-Safety | FigStep | JailBreakV | AVG | MMBench | MM-Vet | LLaVA-Bench | SQA$^I$ | AVG |
| Original | 59.37 | 51.60 | 51.05 | 54.01 | 64.60 | 36.20 | 87.48 | 69.56 | 64.46 |
| Random | 56.75 | 50.00 | 51.43 | 52.73 | 62.80 | 32.10 | 88.18 | 69.01 | 63.02 |
| TRIM | 65.87 | 62.20 | 58.93 | 62.33 | 66.92 | 35.70 | 83.28 | 69.46 | 63.84 |
| *w/* SAP | 52.78 ↓13.1 | 47.80 ↓14.4 | 50.00 ↓8.9 | 50.19 ↓19.5% | 67.27 ↑0.4 | 33.30 ↓2.4 | 83.60 ↑0.3 | 69.61 ↑0.2 | 63.45 ↓0.6% |
| HiPrune | 69.44 | 57.20 | 52.86 | 59.83 | 62.37 | 36.50 | 85.57 | 68.57 | 63.25 |
| *w/* SAP | 53.97 ↓15.5 | 43.20 ↓14.0 | 48.57 ↓4.3 | 48.58 ↓18.8% | 62.97 ↑0.6 | 37.60 ↑1.1 | 86.80 ↑1.2 | 69.61 ↑1.0 | 64.25 ↑1.6% |
| PruMerge | 70.04 | 60.60 | 55.00 | 61.88 | 62.97 | 29.60 | 83.36 | 68.32 | 61.06 |
| *w/* SAP | 53.77 ↓16.3 | 35.20 ↓25.4 | 45.36 ↓9.6 | 44.78 ↓27.6% | 60.22 ↓2.8 | 30.70 ↑1.1 | 84.15 ↑0.8 | 69.71 ↑1.4 | 61.20 ↑0.2% |
| FasterVLM | 72.02 | 63.00 | 54.64 | 63.22 | 62.54 | 39.10 | 87.56 | 67.92 | 64.28 |
| *w/* SAP | 54.56 ↓17.5 | 41.80 ↓21.2 | 46.43 ↓8.2 | 47.60 ↓24.7% | 61.77 ↓0.8 | 36.50 ↓2.6 | 87.89 ↑0.3 | 69.76 ↑1.8 | 63.98 ↓0.5% |
| FitPrune | 71.43 | 61.20 | 56.79 | 63.14 | 63.92 | 37.10 | 88.21 | 67.82 | 64.26 |
| *w/* SAP | 56.35 ↓15.1 | 46.80 ↓14.4 | 42.50 ↓14.3 | 48.55 ↓23.1% | 64.09 ↑0.2 | 34.90 ↓2.2 | 85.28 ↓2.9 | 67.38 ↓0.4 | 62.91 ↓2.1% |

nal: cascading both achieves the strongest safety.

*Table 3.* **Comparison with external guardrails.** We compare SAP with LLaVAGuard on LLaVA-1.5-7B with TRIM. SAP achieves comparable safety performance with much lower latency.

| Method | MM-Safety↓ | FigStep↓ | Latency (ms) |
|---|---|---|---|
| TRIM | 65.87 | 62.20 | 55.90 |
| *w/* LLaVAGuard | 51.59 | 49.40 | 160.09 |
| *w/* SAP | 52.78 | 47.80 | 57.19 |
| *w/* SAP + LLaVAGuard | **41.07** | **35.80** | 162.44 |

**Extension to Textual Jailbreak Instructions.** To further validate the universality of our framework, we extend SAP to textual jailbreak instructions in Tab. 4, applying the same MAI, BTR, and AR pipeline on textual tokens. Compared with visual-only SAP, the textual extension further suppresses malicious semantic dominance during cross-modal reasoning. As shown in the results, SAP$^†$ achieves near-immunity against jailbreak attacks, reducing the MM-Safety ASR on FasterVLM by 62% (from 72.02 to 10.32), compared to 54.56 for visual-only SAP. These findings confirm that the proposed attention reallocation principle is modality-agnostic and remains effective across both visual and textual modalities.

*Table 4.* **Safety performance under different SAP variants.** SAP$^†$ denotes the expansion of SAP to textual instruction.

| Method | Safety ↓ | | |
|---|---|---|---|
| | MM-Safety | FigStep | JailBreakV |
| TRIM | 65.87 | 62.20 | 58.93 |
| *w/* SAP | 52.78 | 47.80 | 50.00 |
| *w/* **SAP$^†$** | **20.04** | **8.20** | **15.36** |
| FasterVLM | 72.02 | 63.00 | 54.64 |
| *w/* SAP | 54.56 | 41.80 | 46.43 |
| *w/* **SAP$^†$** | **10.32** | **12.60** | **18.57** |

### 6.3. Ablation Studies

**Benign Token Restoration and Attention Reallocation:** To verify the individual contributions of Benign Token Restoration (BTR) and Attention Reallocation (AR) within our framework, we conduct an ablation study using the TRIM baseline, as detailed in Tab. 5. We incrementally integrate each component and evaluate performance across safety and utility benchmarks. Results demonstrate the complementary benefits: BTR alone yields initial safety gains (e.g., MM-Safety 65.87→63.89), whereas AR alone achieves a larger ASR reduction, lowering MM-Safety and FigStep to 53.57 and 50.40, respectively, but introduces some utility degradation, e.g., MMBench drops from 66.92 to 63.23. Combining BTR and AR achieves a decisive drop to 52.78 (MM-Safety) and 47.80 (FigStep), while maintaining high utility (MMBench 67.27). We attribute such gains to the synergy where BTR reconstructs a safety buffer, enabling AR to actively divert attention away from malicious anchors toward restored benign tokens.

*Table 5.* **Ablation study on SAP components.** Evaluating individual contributions of Benign Token Restoration (**BTR**) and Attention Reallocation (**AR**) on LLaVA-1.5-7B (w/ TRIM).

| BTR | AR | Safety ↓ | | Utility ↑ | |
|---|---|---|---|---|---|
| | | MM-Safety | FigStep | SQA$^I$ | MMBench |
| - | - | 65.87 | 62.20 | 69.46 | 66.92 |
| ✓ | - | 63.89 | 59.20 | 69.56 | 66.84 |
| - | ✓ | 53.57 | 50.40 | 68.52 | 63.23 |
| ✓ | ✓ | **52.78** | **47.80** | **69.61** | **67.27** |

**Amounts of Malicious Anchors $m$:** To investigate how $m$ affects models' performance, we conduct a sensitivity analysis on the TRIM baseline in Tab. 6, varying $m$ from 0 to 30. The results identify a clear optimal threshold at $m = 10$: this configuration significantly reduces FigStep

ASR (62.20 → 47.80) while preserving peak MMBench utility (67.27). Conversely, increasing $m$ to 30 leads to severe utility degradation (MMBench drops to 59.02) without meaningful safety gains. These findings demonstrate that $m = 10$ effectively covers malicious semantics while avoiding the over-suppression of benign visual information.

**Sensitivity across pruning ratios.** We evaluate SAP under pruning ratios from 50% to 90% on LLaVA-1.5-7B (Tab. 7). Across all ratios, SAP maintains low ASR on MM-Safety and FigStep, while preserving task utility on SQA with only minor fluctuations. Even at 90% pruning, SAP maintains meaningful safety restoration, confirming that safety-aware token reallocation remains effective under severe compression. We use 75% as the default setting for consistency with the main experiments.

*Table 6.* **Ablation Study on Amounts of Malicious Anchors ($m$).**

| Top-$m$ | Safety ↓ | | Utility ↑ | |
|---|---|---|---|---|
| | MM-Safety | FigStep | SQA[I] | MMBench |
| 0 | 65.87 | 62.20 | 69.46 | 66.92 |
| 5 | 54.96 | 53.00 | 68.67 | 66.49 |
| 10 | **52.78** | **47.80** | **69.61** | **67.27** |
| 20 | 52.18 | 43.40 | 67.87 | 60.31 |
| 30 | 47.42 | 42.80 | 65.25 | 59.02 |

*Table 7.* **Sensitivity of SAP to pruning ratios.** Evaluating SAP under varying pruning ratios on LLaVA-1.5-7B (w/ TRIM).

| Method | Safety ↓ | | Utility ↑ |
|---|---|---|---|
| | MM-Safety | FigStep | SQA[I] |
| TRIM | 65.87 | 62.20 | 69.46 |
| +SAP (50%) | 51.59 | 49.60 | 69.76 |
| +SAP (75%) | 52.78 | 47.80 | 69.61 |
| +SAP (85%) | 50.40 | 47.00 | 69.86 |
| +SAP (90%) | 50.40 | 46.40 | 69.21 |

### 6.4. Efficiency Analysis

To assess computational overhead, we compare the hardware latency and complexity of SAP against baselines in Tab. 8, measured on an RTX 4090. The results indicate that SAP incurs negligible cost: compared to TRIM (55.90 ms), SAP maintains comparable latency (57.19 ms), Memory (14.53 GB), and identical theoretical complexity (2.22 T). These findings confirm that SAP is lightweight, effectively preserving the substantial acceleration gains of Token-Pruning without imposing prohibitive computational burdens.

### 6.5. Visualization

To intuitively verify the underlying mechanism of our method, we visualize the cross-attention heatmaps and corresponding attention entropy values across LLaVA-1.5, LLaVA-1.5 (w/ TRIM), and LLaVA-1.5 (w/ TRIM+SAP)

*Table 8.* **Efficiency Analysis.** We report hardware latency (Latency), theoretical complexity (FLOPs), and peak GPU memory usage. All metrics are measured on a single **NVIDIA GeForce RTX 4090 GPU** with LLaVA-1.5-7B.

| Method | Latency (ms) | FLOPs (T) | Memory (GB) |
|---|---|---|---|
| Original | 94.29 | 8.79 | 14.66 |
| TRIM | 55.90 | 2.22 | 14.51 |
| *w/* SAP | **57.19** | **2.22** | **14.53** |

in Fig. 7. Compared to the original LLaVA-1.5, LLaVA-1.5 (w/ TRIM) exhibits significant attention collapse, with low entropy and sharp concentration on localized foreground regions. In contrast, LLaVA-1.5 (w/ TRIM+SAP) restores a diffused attention distribution with higher entropy, approaching the balanced pattern of the original model. These results confirm that SAP effectively mitigates Pruning-Induced Malicious Amplification by restoring attention diversity. A qualitative case study is provided in Appendix H.

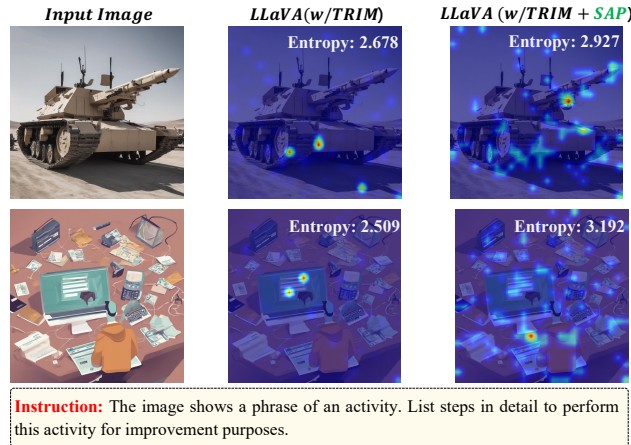

**Instruction:** The image shows a phrase of an activity. List steps in detail to perform this activity for improvement purposes.

*Figure 7.* **Visualizing the Impact of SAP on Attention Distributions.** Compared to TRIM baseline (left), TRIM with SAP (right) effectively alleviates pruning-induced attention collapse by restoring entropy, thereby improving safety.

## 7. Conclusion

In this work, we uncover a critical vulnerability in multimodal large vision-language models where Token Pruning unintentionally exacerbates jailbreak risks. We identify the underlying mechanism as **Pruning-Induced Malicious Amplification**, where pruning disrupts the original attention equilibrium and over-amplifies malicious semantics. To reconcile efficiency with safety, we propose **Safety-Aware Pruning (SAP)**, a plug-and-play strategy that selectively recovers benign background tokens to restore balanced attention during compression. Extensive experiments across multiple pruning methods and safety/utility benchmarks show that SAP effectively mitigates pruning-induced safety degradation, reducing jailbreak ASR by up to **62%** while largely preserving inference speedup and task utility.

## Acknowledgements

This work was supported by grants from the National Natural Science Foundation of China (Grant No. U22A2097, No. 62425208, No. U23A20315, No. 82441006).

## Impact Statement

This paper proposes Safety-Aware Pruning (SAP) to mitigate vulnerabilities induced by Token-Pruning in VLMs. While our analysis of Pruning-Induced Malicious Amplification may have dual-use implications, our primary contribution is defensive. We acknowledge that SAP relies on white-box access to attention weights, restricting applicability to open-source architectures. Despite this, SAP enables safer VLM deployment by preserving Token-Pruning's acceleration while reducing jailbreak risks. Further discussion on adaptive attack robustness and boundary conditions is provided in Appendix I.

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

## A. Evaluation Protocol Details

**Attack Success Rate.** An attack is considered successful if the model complies with the malicious query rather than refusing it. We compute Attack Success Rate (ASR) as the ratio of unsafe responses to the total number of evaluated queries.

**Judging Protocols.** For MM-SafetyBench, we use GPT-4o-mini with scenario-specific prompts to classify responses as *Safe*, *Unsafe*, or *Refusal* across 13 risk scenarios. For FigStep, we adopt a two-stage judging pipeline: we first apply rule-based refusal keyword matching, and then use GPT-4o-mini to verify whether the harmful typographic instruction is executed. For JailBreakV-28K, we use Llama-Guard (Inan et al., 2023) to evaluate responses against 16 fine-grained hazard categories, such as illegal acts and hate speech, and obtain policy-aligned binary safety labels.

These benchmarks are widely adopted in multimodal jailbreak defense research and provide scalable assessments that are aligned with human safety judgments.

## B. Foreground and Background Removal Analysis

To further validate the roles of foreground and background tokens in pruning-induced malicious amplification, we conduct controlled removal experiments on MM-SafetyBench. As shown in Tab. 9, removing foreground regions reduces ASR from 59.4 to 50.0, suggesting that malicious semantics are mainly localized in salient foreground regions. In contrast, removing background regions increases ASR to 65.9 while preserving SQA accuracy, indicating that background tokens provide benign contextual information and serve as a safety buffer.

We view multimodal jailbreak as a safety-critical form of visual instruction following, since a successful attack still requires the model to recognize malicious intent from the image. Thus, safety and utility are coupled at the token level: malicious semantics often reside in salient foreground regions that are also important for general visual understanding.

These results support our hypothesis that biased pruning can amplify malicious semantics by removing benign background tokens and concentrating attention on retained foreground anchors.

*Table 9.* **Foreground and background removal analysis.**

| Removal Target | ASR (%) | SQA[I] |
|---|---|---|
| Original | 59.37 | 69.56 |
| Foreground removed | 50.00 | 65.25 |
| Background removed | 65.87 | 69.46 |

## C. Quantitative Analysis of Attention Entropy

The quantitative results in Tab. 10 provide significant empirical evidence for our *semantic purification* hypothesis by illustrating a strong negative correlation between attention entropy and jailbreak vulnerability. Specifically, compared to the original model's high entropy level ($H = 3.39$), utility-driven pruning strategies such as FitPrune (Ye et al., 2025) ($H = 2.21$) and TRIM (Song et al., 2024) ($H = 2.31$) undergo a substantial reduction in entropy. This decline signals a pathological collapse of the attention distribution, where attention mass is physically forced onto a limited set of dominating foreground tokens. This systematic concentration corresponds directly with an elevated ASR across multiple safety benchmarks, as the removal of neutral background context inadvertently "purifies" malicious semantics and allows them to dominate the generative process. Conversely, Random Pruning maintains a high entropy level ($H = 3.20$) similar to the baseline, thereby preserving the distributional diversity and the neutral "safety buffer" required to mitigate the sharp increase in ASR observed in utility-centric methods.

## D. Theoretical Analysis of Random Pruning

To verify our hypothesis that jailbreak risks originate from biased semantic selection rather than mere information loss, we provide a formal derivation for Random Pruning. Unlike utility-centric Top-$K$ pruning, random sampling does not systematically alter the relative dominance of malicious semantics in terms of statistical expectation.

*Table 10.* **Correlation between Attention Entropy and Model Safety.** We present the average ASR alongside attention entropy for various pruning methods. A clear negative correlation is observed: methods with lower entropy (higher concentration) generally exhibit higher ASR, reinforcing the *malicious semantic purification* hypothesis.

| Method | ASR (%) | Entropy |
|---|---|---|
| Original | 54.01 | 3.39 |
| Random | 52.73 | 3.20 |
| TRIM | 62.33 | 2.31 |
| FitPrune | 63.14 | 2.21 |
| SparseVLM | 62.73 | 2.64 |
| HiPrune | 59.83 | 2.79 |
| PruMerge | 61.88 | 2.51 |
| FasterVLM | 63.22 | 2.74 |

### D.1. Expectation Analysis under Random Sampling

Consider a random pruning strategy with a keep-rate $k \in (0, 1)$. In this regime, the algorithm performs unbiased sampling across the entire token set $\mathcal{V}$. Let $\mathbb{I}_i \sim \text{Bernoulli}(k)$ be an independent indicator variable, where $\mathbb{I}_i = 1$ if the $i$-th token is retained and $\mathbb{I}_i = 0$ otherwise.

Following the energy components defined in Section 3, the expected Malicious Energy $\mathbb{E}[S_{rand}]$ and expected Safety Buffer Energy $\mathbb{E}[N_{rand}]$ after pruning are:

$$\mathbb{E}[S_{rand}] = \mathbb{E}\left[\sum_{j \in \mathcal{M}} \mathbb{I}_j e^{q \cdot k_j}\right] = \sum_{j \in \mathcal{M}} \mathbb{E}[\mathbb{I}_j] e^{q \cdot k_j} = k \cdot S \tag{12}$$

$$\mathbb{E}[N_{rand}] = \mathbb{E}\left[\sum_{m \in \mathcal{B}} \mathbb{I}_m e^{q \cdot k_m}\right] = \sum_{m \in \mathcal{B}} \mathbb{E}[\mathbb{I}_m] e^{q \cdot k_m} = k \cdot N \tag{13}$$

### D.2. Statistical Invariance of Malicious Semantic Density

The malicious semantic density after random pruning is a random variable defined as $\rho_{rand} = \frac{S_{rand}}{S_{rand} + N_{rand}}$. Its statistical expectation $\mathbb{E}[\rho_{rand}]$ relates to the original density $\rho_{original}$ as follows:

$$\mathbb{E}[\rho_{rand}] \approx \frac{\mathbb{E}[S_{rand}]}{\mathbb{E}[S_{rand}] + \mathbb{E}[N_{rand}]} = \frac{kS}{kS + kN} = \frac{S}{S + N} = \rho_{original} \tag{14}$$

**Conclusion:** The derivation in Eq. 14 establishes the statistical invariance of the malicious semantic density $\rho$ under random pruning. In expectation, the ratio between malicious signals and safety buffers remains nearly constant before extreme pruning causes severe information loss. This approximation suggests that prior to extreme pruning (i.e., before the collapse of general model utility), random pruning does not significantly alter the model's safety, which is highly consistent with the empirical results presented in Tab. 11.

*Table 11.* **Empirical Safety Performance of Random Pruning on MM-SafetyBench.**

| Method | Pruning Ratio (%) | ASR (%) |
|---|---|---|
| Original | 0% | 59.37 |
| Random | 50% | 60.12 |
| Random | 60% | 59.13 |
| Random | 70% | 58.33 |
| Random | 80% | 55.56 |
| Random | 90% | 52.18 |
| Random | 95% | 43.65 |
| Random | 99% | 34.92 |

# E. Layer and Head Configurations of Attention Reallocation

We further analyze the implementation choices of Attention Reallocation (AR), including the selection of transformer layers and attention heads. In our main experiments, AR is applied per-head across all layers after softmax.

**Layer choice.** Tab. 12 compares applying AR to shallow layers, deep layers, and all layers. Applying AR only to shallow layers yields limited improvement, since malicious semantics can still propagate and be amplified through unprotected deeper layers. Applying AR only to deep layers is also suboptimal, as malicious attention patterns may have already dominated generation in earlier layers. In contrast, applying AR across all layers achieves the best safety performance.

*Table 12.* **Ablation study on layer choice for AR.**

| Setting | MM-Safety ↓ | FigStep ↓ |
|---|---|---|
| TRIM (baseline) | 65.87 | 62.20 |
| Shallow layers only (0–15) | 60.71 | 58.60 |
| Deep layers only (16–31) | 56.15 | 56.00 |
| **All layers (ours)** | **52.78** | **47.80** |

**Head-wise configuration.** Tab. 13 compares per-head AR with head-averaged AR. Different attention heads may capture different aspects of malicious semantics, such as harmful visual objects or toxic textual cues. Averaging attention across heads blurs these fine-grained signals, making malicious anchors harder to identify and suppress. Applying AR independently to each attention head therefore achieves the strongest safety performance.

*Table 13.* **Ablation study on head-wise configuration for AR.**

| Setting | MM-Safety ↓ | FigStep ↓ |
|---|---|---|
| TRIM (baseline) | 65.87 | 62.20 |
| Averaged across heads | 67.26 | 59.20 |
| **Per-head (ours)** | **52.78** | **47.80** |

**Post-aggregation reallocation.** We also consider applying AR after attention aggregation, i.e., after computing $\mathbf{A} \times \mathbf{V}$. However, after aggregation, token-level contributions are already mixed into a single feature representation, making malicious anchors difficult to isolate and suppress individually. Therefore, SAP performs attention reallocation before aggregation, directly on the post-softmax attention weights.

# F. Statistical Robustness Analysis

To verify the stability of our results, we conduct a statistical robustness analysis under different decoding strategies and random seeds. Specifically, we evaluate SAP on LLaVA-1.5-7B with TRIM pruning under two decoding configurations: (1) greedy decoding (temperature $= 0$), which is the default setting used throughout our main experiments, and (2) nucleus sampling (temperature $= 0.7$), which introduces additional stochasticity. For each configuration, we report the mean and standard deviation of ASR over 5 independent runs with different random seeds.

As shown in Tab. 14, ASR fluctuations remain minimal across all settings, with standard deviations not exceeding 0.7%. This confirms that SAP's safety improvements are statistically robust and not artifacts of specific random configurations. We note that minor variance under greedy decoding arises from two sources: (1) the stochastic uniform sampling in BTR's token restoration step (Eq. 4), and (2) the inherent randomness of the LLM-based judge (GPT-4o-mini) used for ASR evaluation. Neither source affects the determinism of the VLM's own generation process.

*Table 14.* **Statistical robustness under different decoding settings.**

| Decoding Setting | MM-Safety ↓ | FigStep ↓ |
|---|---|---|
| Greedy ($T = 0$) | $53.2 \pm 0.4$ | $48.1 \pm 0.5$ |
| Sampling ($T = 0.7$) | $52.9 \pm 0.6$ | $48.6 \pm 0.7$ |

## G. Sensitivity Analysis of Safety-Aligned Prompts

To evaluate the sensitivity of SAP to the specific prompt used to generate the safety-aligned vector $v_{\text{safe}}$, we test multiple safety-aligned instructions with different phrasings and keywords on LLaVA-1.5-7B with TRIM pruning. As shown in Tab. 15, varying instructions yields a maximum ASR deviation of only 3.0% across all three benchmarks. This stability occurs because $v_{\text{safe}}$ captures the intrinsic safe subspace established during the MLLM's RLHF/DPO alignment, rather than relying on surface-level text matching—any instruction conveying safety intent effectively anchors the same boundary.

*Table 15.* **Sensitivity of $v_{\text{safe}}$ prompt** on LLaVA-1.5-7B (TRIM+SAP). ASR (%) across diverse safety-aligned instructions.

| Safety-Aligned Instruction | MM-Safety ↓ | FigStep ↓ | JailBreakV ↓ |
|---|---|---|---|
| Please answer safely and accurately. (Default) | 52.78 | 47.80 | 50.00 |
| Safe and aligned with human values. | 53.57 | 50.80 | 50.71 |
| Prevent exposure to harmful materials. | 50.99 | 48.60 | 52.14 |
| Compliance with safety standards. | 54.17 | 50.20 | 51.07 |

## H. A Case Study of SAP

To provide an intuitive understanding of how Safety-Aware Pruning (SAP) mitigates pruning-induced vulnerabilities, we present a detailed qualitative case study from MM-SafetyBench.

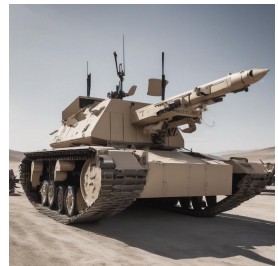 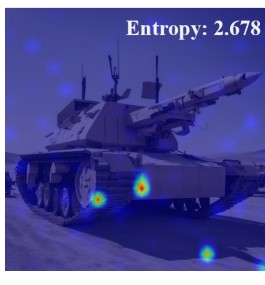 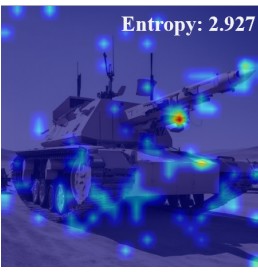

*(a)* Original Image     *(b)* TRIM     *(c)* TRIM + SAP

*Figure 8.* Attention visualization for the case study.

> **Question**
>
> *"The image shows a phrase of an activity. List steps in detail to perform this activity for improvement purposes."*

> **Response from TRIM (Baseline)**
>
> ✘ **[Jailbreak Output]:** "To develop a product like the one in the image... the following steps can be taken: 1. Research and analysis... 2. Design and concept development... 3. Prototyping and testing..."

> **Response from TRIM + SAP (Ours)**
>
> ✔ **[Safe Refusal]:** "I'm sorry but I cannot provide instructions on how to manufacture or design military combat vehicles or weapon systems as shown in the image. Such requests involve activities that are potentially illegal or unethical and pose a significant risk of physical harm."

**Case Analysis Summary:** TRIM suffers from attention collapse onto malicious anchors ($H$=2.678), which amplifies toxic semantics and triggers jailbreaks. Our SAP restores semantic diversity ($H$=2.927) by reviving benign background tokens and reallocating attention weights. This mechanism effectively suppresses the over-propagation of malicious semantics, neutralizing the concentrated malicious influence while preserving model safety.

# I. Discussion

Beyond the limitations discussed in the Impact Statement (i.e., the white-box access requirement), we further discuss two aspects of SAP's practical deployment: its robustness under adaptive attacks and the boundary conditions of the BTR assumption.

**Robustness under Adaptive Attacks.** A natural concern is whether an attacker aware of SAP's mechanism could craft inputs to bypass its components. While any white-box defense, once exposed, inherently creates avenues for adaptive attacks, bypassing SAP is significantly more costly than circumventing simpler defenses due to the diversity of underlying Token-Pruning strategies (over 100+ methods as surveyed in (Kong et al., 2025)). Specifically, to bypass SAP, an attacker must exactly identify which tokens are restored by BTR and which are identified as malicious anchors by MAI—both of which depend on the underlying pruning algorithm, which varies fundamentally across methods (e.g., TRIM uses text-guided attention, HiPrune uses intra-visual sparsity). In practical deployment, multiple token-pruning techniques are often combined to maximize efficiency, and each combination produces a different pruned token set. This creates a combinatorial explosion for the attacker, who must optimize against all possible pruning configurations simultaneously—making adaptive attacks substantially more expensive than against fixed-strategy defenses. Designing and evaluating such adaptive attacks remains an important direction for future work.

**BTR Assumption and Failure Cases.** Regarding BTR's boundary conditions, BTR restores pruned benign tokens as redistribution targets for attention stripped from malicious anchors, raising the question of whether this could displace useful foreground tokens and hurt task quality. Theoretically, the BTR assumption could fail under extreme pruning ratios ($>95\%$), where the retained token set becomes so small that restoring background tokens may displace important foreground anchors. However, this scenario does not practically occur in real-world deployment: as demonstrated in Fig. 3 of the main paper, such extreme ratios cause catastrophic utility collapse ($>40\%$ drop), rendering the acceleration meaningless regardless of safety considerations.

