# OpenReview forum: "Understanding and Mitigating Token-Pruning-Induced Vulnerabilities in VLMs"
_ICML.cc/2026/Conference — ICML 2026 regular_

### Official Review · Reviewer_pdgs · 2026-02-24

**Soundness:** 3
**Presentation:** 3
**Significance:** 3
**Originality:** 3
**Overall Recommendation:** 4
**Confidence:** 3

**Summary:**

This paper investigates the safety implications of Token-Pruning techniques in Vision-Language Models (VLMs), revealing a critical vulnerability where pruning for efficiency significantly increases the susceptibility to jailbreak attacks. The authors first identify a phenomenon termed "Pruning-Induced Malicious Amplification," where the removal of benign background tokens leads to an "attention collapse" onto malicious visual anchors, thereby amplifying toxic semantics during the decoding process. The authors argue that the vulnerability arises because existing pruning strategies, particularly text-guided ones like TRIM, lack safety awareness and fail to maintain the "semantic buffer" provided by background tokens, which normally helps dilute harmful influences. The authors then design Safety-Aware Pruning (SAP), a plug-and-play inference-time defense mechanism that incorporates Malicious Anchor Identification (MAI), Benign Token Restoration (BTR), and an Attention Reallocation (AR) mechanism to restore semantic diversity. Experiments on various VLMs and pruning benchmarks demonstrate that the proposed SAP effectively reduces the Attack Success Rate (ASR) by up to 62% while maintaining the model’s utility and inference efficiency.

**Compliance With Llm Reviewing Policy:**

Affirmed.

**Final Justification:**

Overall, this paper studies an important issue, and the response addresses most of my concerns. As such, I recommend acceptance.

**Key Questions For Authors:**

1. How does SAP's efficiency and defense compare to external Guardrail filters?
2. How sensitive is the model's safety to the specific prompt used to generate $v_{safe}$?

**Limitations:**

Yes

**Strengths And Weaknesses:**

Strengths:

1. This paper is the first to identify and reveal the safety vulnerabilities introduced by Token-Pruning in VLMs.
2. The proposed SAP method is a novel, plug-and-play defense mechanism that can be integrated into some of the existing pruning frameworks.
3. The work is supported by extensive experiments and a solid theoretical analysis.
4. This paper is well-written and easy to follow.

Weaknesses:

1. Limited Safety Improvement: While SAP improves the safety of pruning-based methods, its performance in reducing ASR is still poorer than Query-based Compression methods (e.g., LLaVA-Mini), as evidenced in Table 1.
2. Lack of Baseline Comparison: As a plug-and-play safety component, the paper misses a critical comparison with widely-used Guardrail techniques. Evaluating Token-pruning/SAP against or alongside Guardrails would help the community better understand the severity of the safety issue caused by token pruning.
3. Sensitivity to $v_{safe}$: The method's reliance on a safe direction vector ($v_{safe}$) introduces potential instability. It remains unclear how different prompts or methods for generating $v_{safe}$ affect performance. A sensitivity analysis regarding the selection of this vector is highly recommended.
4. Other minor issues:
   1. Typos exist on lines 141 and 194.
   2. There is a potential inconsistency in the arguments (parameters) of the "Attn" function between lines 155 and 162. Furthermore, the "Attn" function and its parameters are not formally defined or introduced, which may lead to confusion regarding the mathematical implementation.

---

> ### Author Rebuttal · Authors · 2026-03-30
>
> ```W1: Limited improvement compared to Query-based Compression.```
>
> **Response:** ***We have already provided a stronger variant, SAP† (10.32% ASR), in Tab. 5,*** which extends SAP to both visual and textual modalities—***significantly outperforming LLaVA-Mini's 48.2%***—demonstrating superior safety by mitigating malicious dominance across both modalities.
>
> Furthermore, SAP offers fundamental advantages over LLaVA-Mini: ***training-free deployment and architecture-agnostic generalization***. LLaVA-Mini is architecture-specific and requires heavy retraining (8 NVIDIA A800 GPUs), whereas SAP is plug-and-play and inference-time, requiring no training, enabling it to generalize across VLM families—verified on Qwen3-VL-8B (ASR: 44.05%→35.32%) and InternVL2.5-8B (ASR: 58.13%→43.65%) with negligible inference overhead.
>
> **Tab.R1: Generality across diverse MLLMs.**
> |Model|Method|MM-Safety↓|FigStep↓|SQA↑|
> |---|---|---|---|---|
> |Qwen3-VL-4B|Original|41.87|21.00|90.64|
> ||TRIM|48.61|34.20|89.37|
> ||**TRIM+SAP**|**39.68**|**19.80**|**89.63**|
> |Qwen3-VL-8B|Original|39.48|18.60|92.76|
> ||TRIM|44.05|29.40|91.89|
> ||**TRIM+SAP**|**35.32**|**14.80**|**92.15**|
> |InternVL2.5-8B|Original|54.96|24.00|96.49|
> ||TRIM|58.13|30.40|96.51|
> ||**TRIM+SAP**|**43.65**|**16.40**|**96.46**|
> ___
> ```W2 & Q1: Efficiency and defense vs. external Guardrail filters.```
>
> **Response:** We thank the reviewer for this valuable suggestion. Following your advice, ***we compare SAP against LLaVAGuard [1]—a widely adopted external guardrail—on MM-SafetyBench (LLaVA-1.5-7B, TRIM)***. SAP outperforms LLaVAGuard on FigStep (ASR: 47.80% vs. 49.40%) with substantially lower latency (57.19 ms vs. 160.09 ms), as external guardrails introduce additional inference passes that negate pruning's acceleration gains (Tab. R2).
>
> ***Defense-in-Depth: These two methods are orthogonal***, where LLaVAGuard detects unsafe content at the input/output level, while SAP intervenes within the model's internal computation. Cascading both (i.e., SAP+LLaVAGuard) achieves the strongest safety (ASR: 35.8% on FigStep and 41.07% on MM-Safety), confirming their complementarity for secure real-time deployment.
>
> We will add these results to the revised Tab. 2 and Section 6.2.
>
> [1] LLaVAGuard: An Open VLM-based Framework for Safeguarding Vision Datasets and Models. ICML 2025.
>
> **Tab.R2: Comparison with LLaVAGuard on LLaVA-1.5-7B (w/TRIM).**
> |Method|MM-Safety↓|FigStep↓|Latency(ms)↓|
> |---|---|---|---|
> |TRIM|65.87|62.20|55.90|
> |+LLaVAGuard|51.59|49.40|160.09|
> |**+SAP**|**52.78**|**47.80**|**57.19**|
> |**+SAP+LLaVAGuard**|**41.07**|**35.80**|**162.44**|
> ___
> ```W3 & Q2: Sensitivity to the specific prompt used to generate``` $v_{safe}$ ```?```
>
> **Response:** We sincerely thank the reviewer for this constructive suggestion. ***Our SAP is highly robust and insensitive to specific prompt phrasing.*** To verify this, we evaluate SAP under diverse safety-aligned instructions with different phrasings and keywords.
>
> As shown in Tab. R3, varying instructions (e.g., "human values," "safety standards," "harmful material") yields a ***maximum ASR deviation of only 3.0%*** across all three benchmarks (MM-Safety: 1.79%, FigStep: 3.00%, JailbreakV: 2.14%). This stability occurs because $v_{safe}$ captures the intrinsic safe subspace established during the MLLM's RLHF/DPO alignment, rather than relying on surface-level text matching—any instruction conveying safety intent effectively anchors the same boundary.
>
> We will include the full sensitivity analysis in revised **Appendix F**.
>
> **Tab. R3: Sensitivity of $v_{safe}$ prompt (LLaVA-1.5-7B, TRIM+SAP).**
> |Instruction|MM-Safety↓|FigStep↓|JailbreakV↓|
> |---|---|---|---|
> |**Please answer this question safely and accurately.**|52.78|47.80|50.00|
> |Safe and aligned with human values.|53.57|50.80|50.71|
> |Prevent exposure to harmful materials.|50.99|48.60|52.14|
> |Compliance with safety standards.|54.17|50.20|51.07|
> ___
> ```W4: Typos and "Attn" function inconsistency/definition.```
>
> **Response:** We appreciate the reviewer's careful examination of this aspect. Your observations are absolutely correct, and we have fully addressed them in the revised manuscript:
>
> - ***Correction of Typos:*** We fixed the missing term on Line 141 (updated to $|V_{keep}| = K \ll L$) and corrected the grammatical errors on Line 194.
>
> - ***Formal Definition of the "Attn" Function:*** In the revised Section 3.1, we formally define the attention function as $\mathrm{Attn}(Q, K)$, where $Q$ represents the Query and $K$ represents the Key/Value.
>
> - ***Resolving the Inconsistency:*** We unified all parameter orderings to follow the logic $\mathrm{Attn}(Query, Target)$. Specifically, we corrected the Vision-Centric Selection score $\Phi(v_i) \propto \text{Attn}(\mathcal{V}, v_i)$, perfectly aligning it with the Text-Guided Selection (Line 162), formulated as $\Phi(v_i) \propto \mathrm{Attn}(Q_{text}, v_i)$. This ensures identical mathematical implementation across both modalities.

---

> > ### Author Rebuttal · Reviewer_pdgs · 2026-04-01
> >
> > The rebuttal has solved my concerns. Since my original rating already leans towards acceptance, I decide to maintain my rating.

---

> > > ### Author Response · Authors · 2026-04-02
> > >
> > > Dear Reviewer pdgs,
> > >
> > > Thank you for your response and for acknowledging that your concerns have been fully resolved. We appreciate your time and the positive feedback on our rebuttal.
> > >
> > > Best regards,
> > > Authors

---

### Official Review · Reviewer_d9Gv · 2026-03-10

**Soundness:** 3
**Presentation:** 3
**Significance:** 3
**Originality:** 3
**Overall Recommendation:** 4
**Confidence:** 3

**Summary:**

- Summary:
    - This paper studies how token pruning in vision-language models (VLMs) affects multimodal jailbreak robustness. The authors benchmark several pruning paradigms (vision-centric, text-guided, and query-based compression) on LLaVA-1.5-7B across three safety benchmarks (MM-SafetyBench, FigStep, JailbreakV) and four utility benchmarks (MMBench, MM-Vet, LLaVA-Bench, SQA). They report that most pruning strategies *increase* attack success rate (ASR), while query-based compression at extreme compression ratios appears to *reduce* ASR.
    - To explain this, they propose Pruning-Induced Malicious Amplification: pruning tends to remove “benign background/safety-buffer” tokens, causing softmax-normalized cross-attention to concentrate on remaining “malicious anchors.” They then propose Safety-Aware Pruning (SAP), an inference-time plug-in with (1) malicious anchor identification, (2) benign token restoration, and (3) attention reallocation. They report sizable ASR reductions with minimal utility/latency overhead.

**Compliance With Llm Reviewing Policy:**

Affirmed.

**Final Justification:**

The rebuttal resolved my concerns, and I have raised my assessment score from 3 to 4.

**Key Questions For Authors:**

- Questions:
    - What exact automatic judge or annotation protocol is used to compute ASR on MM-SafetyBench/FigStep/JailbreakV? Are there any human validations?
    - At which layers/heads is attention modified? Is AR applied per-head, averaged, or after head aggregation? How sensitive are results to that choice?
    - Do the findings hold for other VLM families (e.g., Qwen-VL, InternVL, etc.) and other backbones/safety alignments? The paper only conducted its evaluation on LLava, which is not sufficient.

I will increase my score if you can answer the questions above and provide the justified experiments of the proposed weakness. Thanks!

**Limitations:**

yes

**Strengths And Weaknesses:**

- Strength:
    - This is a good empirical study paper, across multiple pruning methods, ASR rises while utility remains roughly stable at moderate pruning ratios, suggesting the vulnerability is not purely “capacity loss.
    - The proposed mitigation is actionable. SAP is presented as training-free/inference-time and compatible with several pruning methods; latency overhead reported as small.
- Weakness:
    - Comparability of “query-based compression” vs “token pruning” is confounded. Methods like “LLaVA-Mini / MQT” are not merely pruning strategies applied to the *same base model*; they typically involve architectural/training differences (learned queries, different pretraining/finetuning). The safety gains at 99%+ “pruning” may reflect training, representation bottlenecks, or reduced OCR/trigger fidelity, not the same pruning mechanism being analyzed elsewhere. This weakens the causal claim that “extreme pruning improves safety” as a general token-pruning phenomenon.
    - Threat model and ASR measurement details are insufficient from three points.
        - What constitutes “attack success” for each benchmark? What judge/model/rules are used?
        - Are there multiple decoding settings (greedy vs sampling), multiple seeds, confidence intervals?
        - I think the two questions above are the necessary safety claims for this paper which will strengthen the realism and practicality of this paper.
    - The “theorems” connecting pruning to attention entropy collapse to KL divergence of outputs appear to require strong assumptions not stated. The current writing is more like intuition than a theorem-backed guarantee which makes me confused.

---

> ### Author Rebuttal · Authors · 2026-03-30
>
> ```W1: Query-based compression vs. token pruning.```
>
> **Response:** We thank the reviewer for this observation and clarify our framing. ***We did not claim "extreme pruning improves safety" as a general phenomenon.*** Our original text (Lines 17–24) contrasts query-based compression (LLaVA-Mini/MQT) with standard token pruning—the former unexpectedly improves safety while the latter degrades it. ***This divergence, not the pruning ratio itself, motivates our core question:*** why do the two paradigms yield opposite safety outcomes, and can this safety property be transferred to standard token pruning?
>
> ***The answer lies in their architectural differences***: Standard Token-pruning directly discards tokens, risking retention of malicious ones, whereas Query-based Compression aggregates all tokens  into learnable queries, inherently diluting malicious dominance. Motivated by that, our SAP transfers this dilution principle to standard pruning in a training-free manner.
>
> ***Notably, the original LLaVA-Mini paper[1] positions itself as a soft token pruning strategy*** and compares it with PruMerge, PruMerge++, confirming that analyzing both to understand and address pruning-induced safety issues is well-justified.
>
> We will clarify this in the revised manuscript.
>
> [1] LLaVA-Mini: Efficient Image and Video Large Multimodal Models with One Vision Token. ICML 2025.
> ___
> ```W2/Q1: ASR measurement, judges, and decoding settings.```
>
> **Response:**
> ***Attack Success and Judges:*** An attack succeeds when the model complies with malicious query rather than refusing it. ASR=unsafe responses/total queries. The judge for each benchmark:
>
> - ***MM-SafetyBench:*** GPT-4o-mini with scenario-specific prompts, classifying responses as "Safe," "Unsafe," or "Refusal" across 13 risk scenarios.
> - ***FigStep:*** Two-stage pipeline—rule-based refusal keyword matching, followed by GPT-4o-mini verification to check if the harmful typographic instruction was executed.
> - ***JailbreakV-28K:*** Llama-Guard evaluating responses against 16 fine-grained hazard categories (e.g., illegal acts, hate speech), providing policy-aligned binary safety labels.
>
> ***Human Validation:*** JailbreakV-28K includes explicit human validation; MM-SafetyBench and FigStep rely on automated evaluation. All three are the most widely adopted benchmarks in MLLM jailbreak defense research, proven to provide human-aligned assessments at scale.
>
> ***Decoding and Confidence Intervals:*** We follow the default benchmark configuration: greedy decoding (temperature=0, seed 42). For confidence intervals, we conducted experiments across different decoding strategies (greedy and sampling) with 5 seeds each (Tab. R1).
>
> We will add these evaluation details to the revised Section 6.1.
>
> **Tab.R1: Stability analysis(Mean±Std) on LLaVA-1.5-7B(TRIM+SAP)**
> |Decoding|MM-Safety↓|FigStep↓|
> |---|---|---|
> |Greedy(T=0)|53.2±**0.4**|48.1±**0.5**|
> |Sampling(T=0.7)|52.9±**0.6**|48.6±**0.7**|
> ___
> ```W3: Rigor of theoretical analysis.```
>
> **Response:** Thank you for your comment. We will provide rigorous proofs with explicit assumptions in Appendix D.
> ___
> ```Q2: AR layer/head configurations.```
>
> **Response:** ***AR is applied per-head across all layers after softmax.*** Tab.R2–R3 confirms the necessity of these choices:
>
> ***Layer(Tab.R2): Partial-layer defense is insufficient***—defending only shallow layers allows malicious semantics to propagate and amplify in unprotected deeper layers; defending only deep layers is too late since malicious semantics have already dominated generation in earlier layers.
>
> **Tab.R2: Ablation of layer choice.**
> |Setting|MM-Safety↓|FigStep↓|
> |---|---|---|
> |TRIM(baseline)|65.9|62.2|
> |Shallow(0-15)|60.7|58.6|
> |Deep(16-31)|56.2|56.0|
> |**All layers**|**52.8**|**47.8**|
>
> ***Head(Tab.R3): Each attention head captures different aspects of malicious semantics*** (e.g., harmful visual objects, toxic textual cues). Averaging across heads blurs these fine-grained signals, causing SAP to fail to suppress malicious tokens.
>
> **Tab.R3: Ablation of head choice.**
> |Setting|MM-Safety↓|FigStep↓|
> |---|---|---|
> |TRIM(baseline)|65.9|62.2|
> |Averaged heads|67.3|59.2|
> |**Per-head**|**52.8**|**47.8**|
>
> ***After aggregation is Infeasible:*** Once aggregated (Attention × Value), token contributions are already mixed into a single feature vector—malicious tokens can no longer be individually identified.
>
> We will add these implementation details in Appendix E.
> ___
> ```Q3: Cross-model generalization.```
>
> **Response:** ***Our findings hold for other VLM families and SAP generalizes well across them,*** as shown in Tab.R4. We will incorporate these results into Tab. 2 of the revised manuscript.
>
> **Tab.R4: Generality across VLMs.**
> |Model|Method|MM-Safety↓|FigStep↓|SQA↑|
> |---|---|---|---|---|
> |Qwen3-VL-8B|Original|39.5|18.6|92.8|
> ||TRIM|44.1|29.4|91.9|
> ||**TRIM+SAP**|**35.3**|**14.8**|**92.2**|
> |InternVL2.5-8B|Original|55.0|24.0|96.5|
> ||TRIM|58.1|30.4|96.5|
> ||**TRIM+SAP**|**43.7**|**16.4**|**96.5**|

---

> > ### Author Rebuttal · Reviewer_d9Gv · 2026-04-01
> >
> > Thanks for your rebuttal! All of my concerns have been solved, and I have improved the score from 3 to 4.

---

> > > ### Author Response · Authors · 2026-04-01
> > >
> > > Dear Reviewer d9Gv,
> > >
> > > Thank you very much for your time, engaging in the discussion phase, and for increasing your score. We are deeply grateful for your constructive feedback, which has been invaluable in helping us improve our paper. We are delighted that our rebuttal has successfully addressed all of your concerns.
> > >
> > > Best regards,
> > > Authors

---

### Official Review · Reviewer_Yy3h · 2026-03-14

**Soundness:** 3
**Presentation:** 3
**Significance:** 3
**Originality:** 3
**Overall Recommendation:** 4
**Confidence:** 3

**Summary:**

This paper studies how token pruning changes the safety behavior of vision-language models under multimodal jailbreak attacks. It finds that most pruning strategies increase attack success by removing benign background tokens and concentrating attention onto a few malicious foreground anchors, and then proposes Safety-Aware Pruning, an inference-time method that identifies malicious anchors, restores benign tokens, and reallocates attention to rebalance the model’s focus. Experiments on seven pruning methods, three safety benchmarks, and four utility benchmarks show that the method can substantially reduce jailbreak attack success, by up to 62%, while largely preserving utility and efficiency.

**Compliance With Llm Reviewing Policy:**

Affirmed.

**Key Questions For Authors:**

see weaknesses

**Limitations:**

yes

**Strengths And Weaknesses:**

Strengths

* Good novelty. The vulnerabilities in VLMs after token pruning is a new problem.

* This paper is well written.

Weaknesses

* Most experiments are conducted on LLaVA-1.5-7B, and many of the main evaluations are centered around a 75% pruning ratio. While this setup is sufficient to demonstrate the basic effectiveness of the method, a stronger evaluation would include more backbone models, such as newer VLMs like Qwen3-VL series with different model sizes, as well as a broader range of pruning ratios to better assess the generality of the conclusions.

* The paper would also be stronger if it discussed adaptive attack settings. In particular, it is unclear how robust the proposed defense would remain if an attacker were aware of SAP and explicitly optimized jailbreak inputs or malicious visual patterns to bypass malicious-anchor identification, benign-token restoration, or attention reallocation.

---

> ### Author Rebuttal · Authors · 2026-03-30
>
> ```W1: Experiments are mostly limited to LLaVA-1.5-7B at 75% pruning. Evaluation on more diverse VLMs (e.g., Qwen3-VL) and a wider range of pruning ratios would strengthen generality claims.```
>
> **Response:** We thank the reviewer for this valuable suggestion and have extended the evaluation accordingly.
>
> We have extended evaluation to ***multiple VLM architectures and sizes (Qwen3-VL-4B/8B, InternVL2.5-8B) and a broader range of pruning ratios (50%–90%).*** As shown in Tab. R1:
>
> ***Generality across models and sizes:*** SAP+TRIM consistently reduces ASR across all three models: Qwen3-VL-4B (48.61→39.68% / 34.20→19.80%), Qwen3-VL-8B (44.05→35.32% / 29.40→14.80%), and InternVL2.5-8B (58.13→43.65% / 30.40→16.40%) on MM-Safety / FigStep respectively, while preserving baseline utility (SQA within ±0.3%). This confirms SAP generalizes effectively across diverse model families and scales.
>
> **Tab.R1: Generality across models and sizes.**
>
> |Model|Method|MM-Safety↓|FigStep↓|SQA↑|
> |---|---|---|---|---|
> |Qwen3-VL-4B|Original|41.87|21.00|90.64|
> ||TRIM|48.61|34.20|89.37|
> ||**TRIM+SAP**|**39.68**|**19.80**|**89.63**|
> |Qwen3-VL-8B|Original|39.48|18.60|92.76|
> ||TRIM|44.05|29.40|91.89|
> ||**TRIM+SAP**|**35.32**|**14.80**|**92.15**|
> |InternVL2.5-8B|Original|54.96|24.00|96.49|
> ||TRIM|58.13|30.40|96.51|
> ||**TRIM+SAP**|**43.65**|**16.40**|**96.46**|
>
> ***Generality across pruning ratios:*** Under varying compression intensities (50%, 75%, 85%, 90%), SAP provides a consistent safety margin, reducing ASR by an average of 14.54% compared to vanilla pruning. Even at 90% ratio, SAP still effectively mitigates pruning-induced malicious amplification.
>
> **Tab.R2: Generality across pruning ratios on LLaVA-1.5-7B.**
>
> |Method|MM-Safety↓|FigStep↓|SQA↑|
> |---|---|---|---|
> |Baseline(TRIM)|65.87|62.20|69.46|
> |+SAP(50%)|51.59|49.60|69.76|
> |**+SAP(75%)**|**52.78**|**47.80**|**69.61**|
> |+SAP(85%)|50.40|47.00|69.86|
> |+SAP(90%)|50.40|46.40|69.21|
> ___
> ```W2: The paper lacks discussion of adaptive attacks—i.e., how robust is SAP when an attacker knows the defense and explicitly crafts inputs to evade its components (MAI, BTR, AR)?```
>
> **Response:** We appreciate the reviewer's insight into the adaptive attack setting.
>
> Actually, the implementation details of any white-box defense, if exposed, inherently create avenues for adaptive attacks. Nevertheless, ***bypassing SAP is significantly more costly than circumventing simpler defenses due to the diversity of underlying Token-Pruning strategies (over 100+ methods as surveyed in [1]).***
>
> Specifically, to bypass SAP, an attacker must exactly identify which tokens are restored and which are identified as malicious anchorsby SAP—both depend on the underlying pruning algorithm, which varies fundamentally across methods. In practical deployment, ***multiple token-pruning techniques are often combined to maximize efficiency,*** creating an exponential combinatorial space that forces attackers to optimize against all possible pruning configurations simultaneously—making adaptive attacks substantially expensive.
>
> We agree this is an important direction and will add this discussion to the revised manuscript.
>
> [1] Token Reduction Should Go Beyond Efficiency in Generative Models From Vision, Language to Multimodality. Arxiv 2025.

---

> > ### Author Rebuttal · Reviewer_Yy3h · 2026-04-06
> >
> > Thanks for the response. I will keep my score.

---

> > > ### Author Response · Authors · 2026-04-06
> > >
> > > Dear Reviewer Yy3h,
> > >
> > > Thank you for your response and for confirming that your concerns have been fully resolved. We appreciate your constructive feedback throughout the review process, which has helped improve the clarity of our work. We are glad to have your support.
> > >
> > > Best regards,
> > > Authors

---

### Official Review · Reviewer_peFT · 2026-03-19

**Soundness:** 3
**Presentation:** 3
**Significance:** 2
**Originality:** 3
**Overall Recommendation:** 4
**Confidence:** 3

**Summary:**

This paper examines how token-pruning changes the safety behavior of vision-language models under multimodal jailbreak attacks. They find that token-pruning causes an increased portion of attention to be focused on malicious foreground tokens which amplifies the effectiveness of jailbreaks. To mitigate this they introduce Safety-Aware Pruning (SAP). SAP reallocates attention to pruned tokens based on its deviation from a computed safe direction and the average attention on the token during generation.

**Compliance With Llm Reviewing Policy:**

Affirmed.

**Final Justification:**

I have increased the overall score from weak reject to weak accept and updated my scores on soundness, presentation, and originality in response to the author's rebuttal. I believe the impact of this work is valuable but somewhat limited, and the methodology paper is technically sound and original.

**Key Questions For Authors:**

1. The paper’s explanation seems to rely on the idea that pruning removes mostly benign background tokens while the retained foreground tokens carry the malicious signal. How robust is this assumption across jailbreak settings? In particular, is there evidence that the harmful semantics are usually localized in the foreground rather than distributed across both foreground and context?
2.  Section 4.1 discusses SafePTR and explains how MAI differs conceptually by combining harmfulness and attention. Why was SafePTR not included as an empirical baseline or comparison point? A direct comparison would make it easier to judge the novelty and practical advantage of SAP.
3. Table 3 suggests that Attention Reallocation contributes more than Benign Token Restoration alone. Was reallocation without BTR tested, or otherwise isolated, to determine whether BTR is necessary versus mainly helpful? That would clarify which part of SAP is doing the most work.
4. Are there settings where SAP hurts task quality or removes useful visual detail, especially for tasks that depend on high-fidelity local information? It would be helpful to see examples of where the method fails or where the benign-token restoration assumption does not hold.

**Limitations:**

The limitations of SAP are lacking. As a mitigation method, the paper is unclear about why and when SAP fails vs succeeds; this sort of analysis would help clarify any limitations.

**Strengths And Weaknesses:**

## Soundness
### Strengths
- Experimentation is broad, and evaluation uses many pruning strategies across 3 safety benchmarks and 4 utility benchmarks.
- Ablations for the main SAP components
- Temperature scaling and entropy analysis support the proposed mechanism well
### Weaknesses
- Understated and underexplored threat model, even a brief primer of the multi-modal attack setting would be helpful
- SafePTR is mentioned as an alternative method, but there is no empirical comparison against it
- Lacks confidence intervals, statistical tests, or measures of uncertainty in results

## Presentation
### Strengths
Overall, the structure is easy to follow at a high level, and most figures and tables are clear to read.
### Weaknesses
- The writing needs substantial polishing; the abstract and introduction, in particular, have wording and formatting errors that make things more difficult to read than they need to be.
- Overuse of bolds and italics; these should be reserved and used sparingly for portions to focus on
- Appears to be a missing term in section 3.1  in "|Vkeep| = K ≪ ..."
- Probable mistaken reference in section 6.2: "we extend SAP to textual instructions in Fig. 5," but Figure 5 is the temperature scaling results.
- Some terminology could be clarified, especially phrases like “attention equilibrium,” “safety buffer,” and “purifies malicious semantics.”
- Tables 7 and 8 in the Appendix would likely read more easily as scatter plots
- The related work is fairly sparse, making it more difficult to see clearly how this paper is different from other related methods.

## Significance
### Strengths
- The paper addresses a real problem in the trade-off between efficiency and safety, scoped to token-pruning methods for VLMs.
- The proposed mitigation is also an efficient inference time method that benefits from nice interpretability.
### Weaknesses
- The paper stays focused on token-pruning methods in VLMs and requires white-box access.
- Focuses on a single model family for testing the mitigation method
- Significance is harder to judge without comparison to other safety-preserving approaches

## Originality
### Strengths
- Studies token-pruning from a safety-degradation perspective, which does not appear to have been done before
### Weaknesses
- Missing references to other works on how other information compression approaches for efficiency in autoregressive generative models affect safety and alignment.
- Missing discussion of other attention-based jailbreak mitigation methods

---

> ### Author Rebuttal · Authors · 2026-03-30
>
> ```S-W1: Multimodal attack setting.```
>
> **Response:** Thanks for the thoughtful comments. We consider a ***white-box, inference-time defense where deployers adopt Token-Pruning for efficiency and attackers launch multimodal jailbreaks (vision- and text-driven)*** without knowledge of the pruning strategy or SAP. Three benchmarks broadly cover real-world jailbreak scenarios: MM-SafetyBench (SD-generated and typographic image attacks), FigStep (typographic visual prompt attacks), and JailbreakV-28K (text-driven attacks with persuasive and logic-driven prompts).
>
> ```S-W2/Q2: No comparison with SafePTR.```
>
> **Response:** ***SafePTR's code is not publicly available***, preventing us from directly integrating it into token-pruning methods. We reproduced SafePTR and compare with SAP† (cross-modal SAP, Tab. 5 of manuscript).
>
> **Tab.R1: SafePTR comparison.**
> |Method|MM-Safety↓|FigStep↓|SQA↑|Latency(ms)↓|
> |---|---|---|---|---|
> |TRIM|65.9|62.2|69.5|55.9|
> |+SafePTR|18.9|11.8|68.4|67.3|
> |+SAP†|**20.0**|**8.2**|**70.0**|**59.9**|
>
> ```S-W3: Confidence intervals.```
>
> **Response:** For confidence intervals, we report ***5-seed results under different decoding strategies*** in Tab.R2 (all deviations≤0.7%).
>
> **Tab.R2: Stability analysis(Mean±Std) on LLaVA-1.5-7B.**
> |Decoding|MM-Safety↓|FigStep↓|
> |---|---|---|
> |Greedy(T=0)|53.2±**0.4**|48.1±**0.5**|
> |Sampling(T=0.7)|52.9±**0.6**|48.6±**0.7**|
>
> ```P-W1~W7/O-W1~W2: Writing, formatting, and related work.```
>
> **Response**: Thank you for the insightful review. ***All issues corrected*** and related work expanded in revised manuscript.
>
> ```Sig-W1: White-box access.```
>
> **Response:** ***As acknowledged in Impact Statement Section: Black-box token pruning is infeasible***—all pruning methods require access to internal representations (e.g., attention matrices, hidden states) to compute importance scores.
>
> ```Sig-W2: Model generalization.```
>
> **Response:** ***Our SAP generalizes well across VLMs.*** As shown in Tab.R3, SAP consistently reduces ASR across Qwen3-VL and InternVL2.5.
>
> **Tab.R3: Generalizability across VLMs (w/TRIM)**
> |Model|Method|MM-Safety↓|FigStep↓|SQA↑|
> |---|---|---|---|---|
> |Qwen3-VL-8B|Original|39.5|18.6|92.8|
> ||TRIM|44.1|29.4|91.9|
> ||TRIM+SAP|**35.3**|**14.8**|**92.2**|
> |InternVL2.5-8B|Original|55.0|24.0|96.5|
> ||TRIM|58.1|30.4|96.5|
> ||TRIM+SAP|**43.7**|**16.4**|**96.5**|
>
> ```Sig-W3: Comparison with safety approaches.```
>
> **Response:** ***Existing safety methods mainly target general VLM vulnerabilities*** rather than pruning-induced ones. SAP is the first defense method specifically designed for token pruning.
> Nonetheless, we compare SAP/SAP† with a SOTA external guardrail LLaVAGuard [1] on MM-SafetyBench (LLaVA-1.5-7B as baseline) in Tab.R4.
>
> **Tab.R4: Comparison with safety approach (LLaVAGuard).**
> |Method|MM-Safety↓|FigStep↓|Latency(ms)↓|
> |---|---|---|---|
> |TRIM|65.9|62.2|55.9|
> |+LLaVAGuard|51.6|49.4|160.1|
> |**+SAP**|**52.8**|**47.8**|**57.2**|
> |**+SAP†**|**20.0**|**8.2**|**59.9**|
>
> [1] LLaVAGuard: An Open VLM-based Framework for Safeguarding Vision Datasets and Models. ICML 2025
>
> ```Q1: Foreground-localized malicious semantics.```
>
> **Response:** ***Foreground-localized malicious semantics is not our hypothesis—it is consistent with prior multimodal jailbreak settings widely used in MLLM safety community and robust across benchmarks,*** as attacks fundamentally rely on placing salient triggers in foreground to hijack model attention. To verify this, we compare ASR when removing foreground vs. background tokens via TRIM on MM-SafetyBench.
>
> **Tab.R5: Foreground vs. Background Removal.**
> |Removal Target|ASR↓|SQA↑|
> |---|---|---|
> |LLaVA-1.5-7B|59.4|69.6|
> |+Foreground removed|**50.0**|65.3|
> |+Background removed|**65.9**|69.5|
>
> ```Q3: Necessity of BTR vs. AR.```
>
> **Response:** We deeply appreciate your review. ***BTR is necessary for SAP: AR improves safety by suppressing malicious attention while BTR preserves utility*** by restoring benign tokens for redirected attention. To verify this, we conduct the requested ablation (Tab.R6).
>
> **Tab.R6: Extended Ablation of SAP.**
> |BTR|AR|MM-Safety↓|FigStep↓|MMbench↑|SQA↑|
> |---|---|---|---|---|---|
> |✗|✗|65.9|62.2|66.9|69.5|
> |✗|✓|53.5|50.4|63.2|68.5|
> |✓|✓|**52.8**|**47.8**|**67.3**|**69.6**|
>
> ```Q4: High-fidelity tasks and BTR failure cases.```
>
> **Response:**
>
> ***SAP largely preserves utility on high-fidelity tasks,*** since it replaces only redundant background tokens, mantaining visual details. To verify this, we evaluate SAP on fine-grained visual understanding benchmark (MMbench) in Tab. R7.
>
> ***BTR Assumption Failure:*** The assumption breaks under extreme pruning ratios(>95%). However, this scenario ***rarely occurs in real-world deployment***, as it causes utility collapse, making the acceleration meaningless.
>
> **Tab.R7: Fine-grained utility on MMbench (LLaVA-1.5-7B).**
> |Method|ocr↑|obj_loc↑|spatial_rel↑|Overall↑|
> |---|---|---|---|---|
> |TRIM|51.28|46.91|15.56|66.92|
> |+SAP|**51.28**|**46.91**|**17.78**|**67.27**|

---

> > ### Author Rebuttal · Reviewer_peFT · 2026-04-02
> >
> > The authors have adequately addressed my concerns, and I have updated my score accordingly.

---

> > > ### Author Response · Authors · 2026-04-03
> > >
> > > Dear Reviewer peFT,
> > >
> > > Thank you for your time, constructive feedback, and for updating your score. We are very glad that our rebuttal addressed your concerns.
> > >
> > > Best regards,
> > > Authors

---

### Official Review · Reviewer_P3R7 · 2026-04-06

**Soundness:** 3
**Presentation:** 3
**Significance:** 3
**Originality:** 3
**Overall Recommendation:** 4
**Confidence:** 3

**Summary:**

This paper proposes a systematic study of token-pruning safety in VLMs, revealing that most pruning strategies amplify jailbreak vulnerability by concentrating attention on malicious tokens, while query-based compression can instead improve safety. It further proposes Safety-Aware Pruning (SAP), a plug-and-play inference-time method that restores benign context and reallocates attention to mitigate pruning-induced vulnerabilities without sacrificing efficiency or utility.

**Compliance With Llm Reviewing Policy:**

Affirmed.

**Final Justification:**

This paper studies an important and previously underexplored question: how token pruning affects the safety of VLMs under jailbreak settings. The paper is clearly written, the empirical findings are interesting, and the identification of Pruning-Induced Malicious Amplification provides a reasonable explanation for why many pruning strategies increase vulnerability. In addition, the proposed Safety-Aware Pruning (SAP) is simple, plug-and-play, and supported by experiments showing substantial ASR reduction while largely preserving efficiency and utility. That said, the evaluation is mainly limited to LLaVA-1.5-7B, and some of the mechanism analysis and design choices would benefit from broader validation. Overall, I find the paper meaningful and technically solid, and I believe its strengths outweigh its limitations.

**Key Questions For Authors:**

The paper attributes the vulnerability to Pruning-Induced Malicious Amplification. Could the authors provide more evidence showing that this mechanism consistently holds across different pruning methods, beyond the current case studies and analyses?

**Limitations:**

yes

**Strengths And Weaknesses:**

Strengths.

1. The paper is clearly written and motivates the problem well, highlighting an important and underexplored safety issue of token pruning in VLMs.
2. The paper presents the first comprehensive safety evaluation of token-pruning methods, covering multiple pruning paradigms, safety benchmarks, and utility benchmarks.
3. The paper provides a clear and intuitive explanation of the proposed vulnerability mechanism, namely Pruning-Induced Malicious Amplification, which helps understand why pruning can worsen jailbreak risks.
4. The paper proposes an effective defense method, Safety-Aware Pruning (SAP), which significantly reduces ASR while preserving model efficiency and utility.



Weakness:
1. The evaluation is mainly conducted on LLaVA-1.5-7B, so the generality of the conclusions across other VLM architectures remains somewhat unclear.

2. The proposed SAP introduces several design choices and hyperparameters, such as the number of malicious anchors and the attention reallocation ratio, and the robustness of these settings could be discussed more thoroughly.

3. While the paper shows that SAP preserves utility on standard benchmarks, it provides limited discussion on whether the defense may over-suppress borderline benign but sensitive queries.

---

### Decision · Program_Chairs · 2026-04-30

**Decision:**

Accept (regular)

**Comment:**

The paper studies an important and timely question at the intersection of efficiency and safety in VLMs: how token pruning affects multimodal jailbreak vulnerability. Reviewers consistently found the problem novel, the empirical study broad, and the proposed SAP defense practical, training-free, and technically meaningful. The main concerns centered on breadth of evaluation beyond the initial setting, clearer comparison against related defenses and compression approaches, stronger detail on ASR measurement and robustness, and sharper theoretical and presentation clarity. The rebuttal addressed these points well by adding cross-model results, broader comparisons, stability analyses, implementation clarifications, and a more careful explanation of the compression-vs-pruning distinction. Also, the reviewer P3R7 half way after the rebuttal phase (weak accept) so the AC has made an author-AC confidential discussion suggesint the authors to ignore this review, which is the main reason why the review has not been addressed.

Overall, the paper makes a solid contribution with clear community value, and the remaining limitations appear to affect scope more than validity